# Set-point optimization in wind farms to mitigate effects of flow blockage induced by atmospheric gravity waves

Luca Lanzilao and Johan Meyers

Department of Mechanical Engineering, KU Leuven, Celestijnenlaan 300 A, 3001 Leuven, Belgium

**Correspondence:** Luca Lanzilao (luca.lanzilao@kuleuven.be)

**Abstract.** Recently, it has been shown that flow blockage in large wind farms may lift up the top of the boundary layer, thereby triggering atmospheric gravity waves in the inversion layer and in the free atmosphere. These waves impose significant pressure gradients in the boundary layer causing detrimental consequences in terms of farm's efficiency. In the current study, we investigate the idea of controlling the wind farm in order to mitigate the efficiency drop due to wind-farm induced gravity

waves and blockage. The analysis is performed using a fast boundary layer model which divides the vertical structure of the atmosphere into three layers. The wind-farm drag force is applied over the whole wind-farm area in the lowest layer and is directly proportional to the wind-farm thrust set-point distribution. We implement an optimization model in order to derive the thrust-coefficient distribution which maximizes the wind-farm energy extraction. We use a continuous adjoint method to efficiently compute gradients for the optimization algorithm, which is based on a quasi-Newton method. Power gains are

evaluated with respect to a reference thrust-coefficient distribution based on the Betz–Joukowsky set point. We consider thrust coefficients that can change in space, as well as in time, i.e. considering time-periodic signals. However, in all our optimization results, we find that optimal thrust-coefficient distributions are steady; any time-periodic distribution is less optimal. The (steady) optimal thrust-coefficient distribution is inversely related to the vertical displacement of the boundary layer. Hence, it assumes a sinusoidal behaviour in the streamwise direction in subcritical flow conditions, whereas it becomes a U-shaped

curve when the flow is supercritical. The sensitivity of the power gain to the atmospheric state is studied using the developed optimization tool for almost two thousand different atmospheric states. Overall, power gains above 4% were observed for 77% of the cases with peaks up to 14% for weakly stratified atmospheres in critical flow regimes.

## 1  Introduction

Nowadays, it is well known that turbines strongly interact when clustered together in large arrays, increasing the momentum

deficit in the lowest region of the atmospheric boundary-layer (ABL). These turbine–turbine interactions, such as reduced wind speed and increased turbulence intensity, occur within the wind-farm area and can lead to detrimental consequences in terms of farm's efficiency (Barthelmie et al., 2010). However, it has been recently discovered that also non-local effects such as gravity waves may have strong implications on the wind-farm energy extraction (Allaerts and Meyers, 2018, 2019).

In a stable atmosphere, an air parcel which is vertically perturbed will have the tendency to fall back to its original position.

In such case, an oscillation is initiated that is driven by gravity and inertia; this is called a gravity wave. Mountains are examples

of orographic obstacles that trigger vertical flow displacement, and consequently gravity waves (Smith, 1980). The drag force exerted by the mountain is usually transported upward by these waves. At the point of breakdown, the drag force is released in the upper levels of the atmosphere, causing a slow down of the large-scale flow (Eliassen and Palm, 1960; Durran, 1990). Moreover, when air is lifted in a stable atmosphere, a cold anomaly is created, which induces horizontal pressure gradients (Smith, 2010).

In a wind farm, the upward displacement of the boundary layer, caused by diverging fluid streamlines due to flow deceleration by the turbines, can trigger gravity waves in the stable free atmosphere above the boundary layer as well (Smith, 2010; Allaerts and Meyers, 2017). As a result, an adverse pressure gradient develops in the induction region of the wind farm, which slows down the wind-farm inflow velocity (Allaerts and Meyers, 2018). The size of this region scales with the length of the farm. This phenomenon is one possible cause of flow blockage. Note that it differs from classical hydrodynamic blockage caused by the turbine induction, which typically scales with the turbine rotor diameter, and which has also been studied recently in much detail (Bleeg et al., 2018; Segalini and Dahlberg, 2019). The goal of the current study is to determine a wind-farm thrust-coefficient distribution that minimizes the gravity-wave induced blockage effects, maximizing the flow wind speed, and therefore the power production. Moreover, we investigate the impact of different atmospheric conditions on the optimal thrust-coefficient distribution and corresponding power gains.

Gravity waves are related with flows over mountain since a long time (Queney, 1948). However, the cumulated blockage effect induced by the wind farm in the induction region was associated with wind-farm induced gravity waves only in recent years. In the pioneering work of Smith (2010), a quasi-analytical model of atmospheric response to wind-farm drag was used for modelling gravity-wave excitation due to diverging streamlines above the wind-farm area. Results have shown that gravity-wave excitation is strongly dependent upon the height of the boundary layer and the stability of the atmosphere aloft. Later, a fast boundary-layer model was proposed by Allaerts and Meyers (2019) who highlighted the crucial role of the inversion layer in determining gravity-wave patterns. The authors also used this model for an annual energy production study of the Belgian-Dutch offshore wind-farm cluster, showing that the annual energy loss due to the effect of self-induced gravity waves might be on the order of 4 to 6% (Allaerts et al., 2018).

Gravity waves were also observed in mesoscale and Large Eddy Simulation (LES) models. Fitch et al. (2012) and Volker (2014) proposed two different wind-farm parametrizations for the Weather Research and Forecasting model (WRF). Wind-farm induced gravity waves were observed in both cases, causing flow deceleration several kilometers upstream of the farm. Allaerts and Meyers (2017, 2018) have investigated the interaction between an "infinitely" wide wind farm and both a conventionally neutral and stable boundary layer in typical offshore conditions in a LES framework. They found that for low ABL heights, gravity waves induce strong pressure gradients and play an important role in the distribution of the kinetic energy within the farm. Wu and Porté-Agel (2017) considered a large finite-size wind farm operating in a conventionally neutral boundary layer (CNBL) with different free atmosphere stratification, and they conclude that strongly stratified atmospheres decrease the turbine power output up to 35% with respect to the weakly stratified cases. Wind-farm flow blockage was also detected in field measurements. Wind speed data taken before and after the placement of three wind farms showed that there was a reduction in wind speed of about 3% in the induction region of each wind farm after that turbines were installed (Bleeg et al., 2018).

In the last decades, a considerable amount of research has focused on wind-farm control strategies that allow to maximize the farm power output. We refer to Kheirabadi and Nagamune (2019) for a recent comprehensive overview. However, earlier studies all focus on influencing wake dynamics and wake mixing, which occur at a much smaller scale than wind-farm induced gravity waves, to improve power extraction in waked turbines. Important control mechanisms include wake redirection (by yawing and tilting of the turbine), and turbine de-rating strategies. Control actions that influence wind-farm physics on a much large scale, such as self-induced gravity waves, are not explored to date.

In the current work, we concentrate on using wind-farm control to alter/improve the interaction between the wind farm and its self-induced gravity wave system. To this end, we use the fast boundary-layer model proposed by Allaerts and Meyers (2019) which divides the vertical structure of the atmosphere into three layers (from here the name three-layer model) and we reformulate it as an optimization problem. The objective function is defined as the wind-farm energy extracted over a time period $T$, while the constraints are the model equations plus a box constraint for the wind-farm thrust set-point distribution $C_T(x,y,t)$. Note that we do not use the tip-speed ratio and/or the pitch angle distribution as control parameters. Instead, we directly control the thrust set-point distribution. In fact, the former approach would not add further insight in the study performed in the current manuscript. The model equations are derived following the theory for interacting gravity waves and boundary layers developed by Smith et al. (2006), Smith (2007) and Smith (2010). Consequently, the optimal thrust-coefficient distribution computed using the optimization formulation of the three-layer model takes into account the effects of self-induced gravity waves. Hence, we investigate whether it is possible to mitigate gravity-wave induced blockage effects by varying the thrust set-point distribution within the wind-farm area.

The remainder of this paper is formulated as follows. The three-layer model and its optimization formulation is introduced in Section 2. Next, Section 3 describes the numerical setup, wind-farm layout and atmospheric state. Thereafter, Section 4 presents optimization results. The optimal thrust set-point distributions obtained in two different flow cases are discussed in Section 4.1. The sensitivity of the power gain to the atmospheric state is carried out in Section 4.2. Finally, conclusions and suggestions for further research are given in Section 5 .

## 2 Methodology

We now introduce the approach used for modelling wind-farm induced gravity waves and the method applied for maximizing the wind-farm energy output. The three-layer model is described in Section 2.1 and its optimization formulation is derived in Section 2.2.

### 2.1 Three-layer model

In the work of Smith (2010), the atmospheric response to wind-farm drag is simulated by dividing the vertical structure of the atmosphere in two layers: the ABL and the free atmosphere aloft. This approach has strong limitations. In fact, the author is implicitly assuming that the turbine drag is mixed homogeneously between turbine level and the top of the ABL. In real wind farms, the turbine drag slows down the flow only within few hundreds meters from the ground level, triggering the formation

of an internal boundary layer (Wu and Porté-Agel, 2013; Allaerts and Meyers, 2017). To overcome the limitations of Smith's model, the three-layer model divides the ABL into two layers: the wind-farm layer in which the turbine forces are felt directly (a layer's height of twice the turbine hub height has been used by Allaerts and Meyers (2019) based on insights from LES in Allaerts et al. (2018)), and a second layer up to the top of the ABL. Finally, the third layer models the free atmosphere above the ABL following the approach of Smith (2010).

The three-layer model has been validated against LES results by Allaerts and Meyers (2019) (see Section 3 VAL2) on a two dimensional (x-z) domain (i.e., all spanwise derivatives are set to zero). The model shows a mean absolute error (MAE) of 1.3% and 1.8% in terms of maximum displacement of the inversion layer and maximum pressure disturbance, respectively. Moreover, the model underestimates the velocity over the wind-farm area with a MAE of 5.6%. Note that the three-layer model is a linearized model, hence the discrepancies with LES results increase with increasing perturbation values. In fact, the model agrees very well with LES data when perturbations are small (i.e, when non-linear effects are negligible). From this perspective, it may be expected that errors decrease slightly in optimized settings in which perturbation magnitudes are typically lower. For further details about model validation, we refer to Allaerts and Meyers (2019).

The model equations are derived starting from the incompressible three-dimensional Reynolds-Averaged Navier-Stokes (RANS) equations for the ABL (Stull, 1988). A depth-integration over the wind-farm and upper layer height is further computed, which removes the vertical velocity from the equations. Hence, the basic equation system is reduced to a set of only three equations: the continuity equation and the momentum equations in horizontal directions. Subsequently, the governing equations are linearized with respect to the background state variables, using some additional modelling assumptions for the turbulent stresses (see Allaerts and Meyers (2019) for more details). As a result, we are using the following equations for the two layers in the ABL

$$\frac{\partial \boldsymbol{u}_1}{\partial t} + \boldsymbol{U}_1 \cdot \nabla \boldsymbol{u}_1 + \frac{1}{\rho_0} \nabla p + f_c \boldsymbol{J} \cdot \boldsymbol{u}_1 - \nu_{t,1} \nabla^2 \boldsymbol{u}_1 - \frac{\boldsymbol{D}'}{H_1} \cdot (\boldsymbol{u}_2 - \boldsymbol{u}_1) + \frac{\boldsymbol{C}'}{H_1} \cdot \boldsymbol{u}_1 = \frac{\boldsymbol{f}}{H_1}, \tag{1}$$

$$\frac{\partial \eta_1}{\partial t} + \boldsymbol{U}_1 \cdot \nabla \eta_1 + H_1 \nabla \cdot \boldsymbol{u}_1 = 0, \tag{2}$$

$$\frac{\partial \boldsymbol{u}_2}{\partial t} + \boldsymbol{U}_2 \cdot \nabla \boldsymbol{u}_2 + \frac{1}{\rho_0} \nabla p + f_c \boldsymbol{J} \cdot \boldsymbol{u}_2 - \nu_{t,2} \nabla^2 \boldsymbol{u}_2 + \frac{\boldsymbol{D}'}{H_2} \cdot (\boldsymbol{u}_2 - \boldsymbol{u}_1) = 0, \tag{3}$$

$$\frac{\partial \eta_2}{\partial t} + \boldsymbol{U}_2 \cdot \nabla \eta_2 + H_2 \nabla \cdot \boldsymbol{u}_2 = 0. \tag{4}$$

When the wind farm is not operating (i.e., the wind-farm drag force is zero), a horizontally invariant reference state $(\boldsymbol{U}_1, H_1)$ and $(\boldsymbol{U}_2, H_2)$ characterizes the wind-farm and upper layer, where $\boldsymbol{U}_1 = (U_1, V_1)$ and $\boldsymbol{U}_2 = (U_2, V_2)$ are the height-averaged horizontal components of the background velocity and $H_1$, $H_2$ represent the height of the two layers. Whenever the farm extracts power from the flow, small velocity and height perturbations $(\boldsymbol{u}_1, \eta_1)$ and $(\boldsymbol{u}_2, \eta_2)$ are triggered. The equations derived by Allaerts and Meyers (2019) predict the spatial evolution of these perturbations. In this article, we also consider the temporal evolution, and thus, the relevant time derivatives are added to the equations. Furthermore, $\rho_0$ denotes the air density, assumed constant within the ABL, $\nu_{t,1}$ and $\nu_{t,2}$ are the depth-averaged turbulent viscosity, $f_c = 2\Omega \sin \phi$ is the Coriolis frequency, with $\Omega$ the angular velocity of the earth and $\phi$ the latitude, $\boldsymbol{J} = \boldsymbol{e}_x \otimes \boldsymbol{e}_y - \boldsymbol{e}_y \otimes \boldsymbol{e}_x$ is the two-dimensional rotation dyadic with $\boldsymbol{e}_x$ and

$e_y$ two-dimensional unit vectors in the $x$- and $y$-direction, respectively. Finally, the perturbation of the friction at the ground and at the interface between both layers are described by the matrices $C'$ and $D'$.

The right-hand side of Eq. 1 is characterized by the wind-farm drag force $f$. We use a box-function wind-farm force model similar to Smith (2010) in our study. This allows us to avoid the complexity of wake models while gaining in computational time. In fact, this model uniformly spreads the force over the simulation cells in the wind-farm area and does not represent the disturbances caused by each turbine in detail. The force magnitude depends on the wind-farm layout, the wind speed and the thrust set-point distribution (i.e., the $C_T$ value in every grid cell within the farm). As for the flow equations, the wind-farm drag force model is linearized around a constant background state. We retain the first two terms of the Taylor expansion; both scale linearly with the thrust-coefficient distribution. Hence, the drag force is given by $f = f^{(0)} + f^{(1)}$ with

$$f^{(0)} = -\beta C_T B(x,y)\|U_1\|U_1, \tag{5}$$

$$f^{(1)} = -\beta C_T B(x,y)U' \cdot u_1, \tag{6}$$

where

$$U' = \frac{1}{\|U_1\|}\left(U_1 \otimes U_1 + I\|U_1\|^2\right) \tag{7}$$

and with $B(x,y)$ a box function equal to one within the wind-farm area and zero outside. The $x$- and $y$-axis denote the streamwise and spanwise direction, respectively. The wind-farm drag force magnitude in Eqs. 5 and 6 scales with

$$\beta = \frac{\pi \eta_{\mathrm{w}} \gamma}{8 s_x s_y}, \tag{8}$$

where $s_x$ and $s_y$ are the streamwise and spanwise turbine spacing relative to the rotor diameter, $\eta_{\mathrm{w}}$ is the wake efficiency and $\gamma = u_r^2/\|U_1\|^2$ is a velocity shape factor with $u_r$ the rotor-averaged wind speed (Allaerts and Meyers, 2018). Moreover, $I = e_x \otimes e_x + e_y \otimes e_y$ denotes the unit dyadic. Finally, $C_T(x,y,t)$ represents the thrust-coefficient distribution. To compute the thrust coefficient $\widetilde{C}_{T,k}(t)$ of a turbine at location $(x_k, y_k)$, it is possible to evaluate the thrust set-point distribution $C_T(x_k, y_k, t)$. A more accurate connection between $\widetilde{C}_{T,k}(t)$ and the drag force $f$ would, e.g., require the use of an analytical wind-farm wake model. This is however not considered in the current work, so that wake effects are not explicitly incorporated in the optimization. Rather, we consider the optimization of the gravity-wave system, while presuming that the wake efficiency parameter $\eta_{\mathrm{w}}$ does not change as a result of the optimization. Relation 6 is nonlinear since it contains a product between time and space-dependent variables (i.e, $C_T$ and $u_1$). We decide to retain this term because it allows us to include gravity-wave feedback on wind-farm energy extraction. In fact, $f^{(1)} \geq 0$ so that it reduces the drag force that the farm exerts on the flow, thereby reducing effects of blockage in the model. We note that Allaerts and Meyers (2019) have shown that the flow perturbations computed with this simple drag force model have similar trends and orders of magnitude as the ones computed using a drag model that relies on the more detailed analytical wake model of Niayifar and Porté-Agel (2016). Therefore, we believe that the model adopted is a reasonable representation of reality.

The total vertical displacement of the inversion layer $\eta_t = \eta_1 + \eta_2$ triggers gravity waves which induce pressure perturbations $p$. The relation between these two quantities is given by Smith (2010)

$$\frac{p}{\rho_0} = \mathcal{F}^{-1}(\hat{\Phi}) * \eta_t \tag{9}$$

where $\mathcal{F}^{-1}$ and $*$ denote the inverse Fourier transform and the convolution product, respectively. The pressure $p$ is evaluated at the capping inversion height and it is assumed to be constant through the whole ABL (using the classical boundary-layer approximation $\partial p / \partial z = 0$). The complex stratification coefficient $\hat{\Phi}$ in Fourier components is expressed as

$$\hat{\Phi} = g' + \frac{i(N^2 - \Omega^2)}{m}. \tag{10}$$

Relation 10 is obtained from linear three-dimensional, non-rotating, non-hydrostatic gravity-wave theory (Nappo, 2002) under the assumption of constant wind speed $\boldsymbol{U}_g = (U_g, V_g)$ and Brunt-Väisälä frequency $N$. The reduced gravity $g' = g \Delta\theta / \theta_0$ accounts for two-dimensional trapped lee waves (further named as inversion waves) which corrugate the capping inversion layer. The potential-temperature difference $\Delta\theta$ denotes the strength of the capping inversion, and $\theta_0$ is a reference potential temperature. The effect of internal gravity waves is represented by the second term of relation 10, where $m$ denotes the vertical wavenumber which is given by

$$m^2 = (k^2 + l^2)\left(\frac{N^2}{\Omega^2} - 1\right). \tag{11}$$

According to the sign of $m^2$ we can have propagating or evanescent waves. Moreover, $\Omega = \omega - \kappa \cdot \boldsymbol{U}_g$ denotes the intrinsic wave frequency with $\kappa = (k, l)$ the horizontal wavenumber vector.

Finally, combining Eq. 9 with Eq. 2 and Eq. 4, we can write the continuity equations for the wind-farm and upper layer as

$$\frac{1}{\rho_0}\frac{\partial p_1}{\partial t} + \frac{1}{\rho_0}\boldsymbol{U}_1 \cdot \nabla p_1 + H_1 \nabla \cdot \left[\mathcal{F}^{-1}(\hat{\Phi}) * \boldsymbol{u}_1\right] = 0, \tag{12}$$

$$\frac{1}{\rho_0}\frac{\partial p_2}{\partial t} + \frac{1}{\rho_0}\boldsymbol{U}_2 \cdot \nabla p_2 + H_2 \nabla \cdot \left[\mathcal{F}^{-1}(\hat{\Phi}) * \boldsymbol{u}_2\right] = 0 \tag{13}$$

where $p = p_1 + p_2$ is intended as the sum of the pressure perturbations induced by the vertical displacements $\eta_1$ and $\eta_2$, respectively. This form will be used in the remainder of the manuscript.

## 2.2 Optimization model

The goal of the optimization framework is to find a time-periodic optimal thrust-coefficient distribution $C_T^O(x, y, t)$ that minimizes the gravity-wave induced blockage effects, maximizing the flow wind speed and consequently the wind-farm energy extraction over a selected time period $T$. The background atmospheric state is presumed to be steady, which is the reason why we use a time-periodic control (i.e. leading to a moving time average of the optimal control that is steady, and does not lead to end-of-time effects). The wind-farm layout and the atmospheric state are inputs of the optimization model and are detailed in Section 3. Note that the relation between overall wind-farm drag and wind-farm blockage is non-trivial. On the one hand,

increased wind-farm drag leads to increased wind-farm blockage induced by gravity waves. This results from mass conservation and the upward displacement of the free atmosphere. On the other hand, increased wind-farm blockage reduces wind-farm

drag. Thus, the aim of the optimization is to find the optimal balance between these two opposing trends.

By using axial momentum theory (Burton et al., 2001), we find that the power coefficient $C_p(x,y,t)$ depends upon the thrust coefficient according to the following non-linear relationship

$$C_p = \frac{C_T}{2}\left(1 + \sqrt{1 - C_T}\right). \tag{14}$$

The objective function of the optimization model consists in the energy extracted by the farm in the time interval $[0,T]$, hence

it is defined as

$$\mathcal{J}\left(\boldsymbol{\psi}, C_T\right) = -\beta \|\boldsymbol{U}_1\| \int\limits_0^T \iint\limits_{\Omega} C_p B(x,y)\left(\|\boldsymbol{U}_1\|^2 + 3\boldsymbol{U}_1 \cdot \boldsymbol{u}_1\right) \mathrm{d}\boldsymbol{x}\mathrm{d}t \tag{15}$$

where $\Omega = D_x \times D_y$ is the computational domain area. The non-linear relationship between $C_p$ and $C_T$ and the product between control and state variables in Eq. 15 imply that the objective function $\mathcal{J}$ is non-convex.

The wind-farm optimal configuration that maximizes the energy output (note that the objective function is defined with a

minus sign) is then obtained by solving the following non-linear time-periodic PDE-constrained optimization problem

$$\min_{\boldsymbol{\psi}, C_T} \quad \mathcal{J}(\boldsymbol{\psi}, C_T) \tag{16}$$

s.t.

$$\frac{\partial \boldsymbol{u}_1}{\partial t} + \boldsymbol{U}_1 \cdot \nabla \boldsymbol{u}_1 + \frac{1}{\rho_0}\nabla p_1 + \frac{1}{\rho_0}\nabla p_2 + f_c \boldsymbol{J} \cdot \boldsymbol{u}_1 - \nu_{t,1}\nabla^2 \boldsymbol{u}_1 - \frac{\boldsymbol{D}'}{H_1}\cdot\left(\boldsymbol{u}_2 - \boldsymbol{u}_1\right) + \frac{\boldsymbol{C}'}{H_1}\cdot\boldsymbol{u}_1 = \frac{\boldsymbol{f}^{(0)} + \boldsymbol{f}^{(1)}}{H_1} \quad \text{in } \Omega \times (0,T],$$

$$\frac{\partial \boldsymbol{u}_2}{\partial t} + \boldsymbol{U}_2 \cdot \nabla \boldsymbol{u}_2 + \frac{1}{\rho_0}\nabla p_1 + \frac{1}{\rho_0}\nabla p_2 + f_c \boldsymbol{J} \cdot \boldsymbol{u}_2 - \nu_{t,2}\nabla^2 \boldsymbol{u}_2 + \frac{\boldsymbol{D}'}{H_2}\cdot\left(\boldsymbol{u}_2 - \boldsymbol{u}_1\right) = 0 \quad \text{in } \Omega \times (0,T],$$

$$\frac{1}{\rho_0}\frac{\partial p_1}{\partial t} + \frac{1}{\rho_0}\boldsymbol{U}_1 \cdot \nabla p_1 + H_1 \nabla \cdot \left[\mathcal{F}^{-1}(\hat{\Phi}) * \boldsymbol{u}_1\right] = 0 \quad \text{in } \Omega \times (0,T],$$

$$\frac{1}{\rho_0}\frac{\partial p_2}{\partial t} + \frac{1}{\rho_0}\boldsymbol{U}_2 \cdot \nabla p_2 + H_2 \nabla \cdot \left[\mathcal{F}^{-1}(\hat{\Phi}) * \boldsymbol{u}_2\right] = 0 \quad \text{in } \Omega \times (0,T],$$

$$0 \le C_T < 1 \quad \text{in } \Omega \times (0,T],$$

$$C_T(x,y,0) = C_T(x,y,T) \quad \text{in } \Omega.$$

The constraints are the state (or forward) equations presented in the previous paragraph. Since Eq. 14 is defined only for

$C_T \in [0,1)$, we added a box constraint to the optimization model. Moreover, the time-periodicity is imposed by assuming $C_T(x,y,0) = C_T(x,y,T)$. The system state $\boldsymbol{\psi} = \left[u_1, v_1, u_2, v_2, p_1, p_2\right]$ includes the velocity and pressure perturbations in the wind-farm and upper layer, which also define the unknowns of the three-layer model. The control parameters consist of the value of the thrust set-point in each grid cell within the wind-farm area. Hence, the size of the control space is proportional to $N_x^{\mathrm{wf}} N_y^{\mathrm{wf}} N_t$, where $N_t$ represents the number of time steps within the time horizon $T$, while $N_x^{\mathrm{wf}}$ and $N_y^{\mathrm{wf}}$ denote the number

of grid points within the wind-farm area along the x and y-direction, respectively.

It is common practice in PDE-constrained optimization problem to not optimize the cost functional $\mathcal{J}(\psi, C_T)$ directly because such a problem would span both the state and control space. To avoid exploring the entire feasibility region, we require $\psi(C_T)$ to be the solution of the state equations throughout the optimization process. In other words, defining an operator $\mathcal{N}(\psi, C_T)$ that denotes the state equations, we are enforcing $\mathcal{N}(\psi(C_T), C_T) = 0$ during optimization iterations. This technique leads us to a reduced optimization problem with feasibility region given by the control space (De Los Reyes, 2015). The reduced optimization problem is written as

$$\min_{C_T} \quad \widetilde{\mathcal{J}}(C_T) = \mathcal{J}(\psi(C_T), C_T) \tag{17}$$

s.t.

$$0 \le C_T < 1 \qquad\qquad\qquad\qquad\qquad \text{in } \Omega \times (0, T],$$

$$C_T(x, y, 0) = C_T(x, y, T) \qquad\qquad\qquad\qquad \text{in } \Omega$$

where the only remaining constraints are the ones on the control parameters.

The gradient of the reduced objective function $\nabla \widetilde{\mathcal{J}}$ is needed for the solution of the reduced optimization problem. To this end, we use the continuous adjoint method. The adjoint (or backward) equations are given by (see Appendix A2 for detailed derivation)

$$-\frac{\partial \boldsymbol{\zeta}_1}{\partial t} - \boldsymbol{U}_1 \cdot \nabla \boldsymbol{\zeta}_1 + f_c \boldsymbol{J} \cdot \boldsymbol{\zeta}_1 - \nu_{t,1} \nabla^2 \boldsymbol{\zeta}_1 + \frac{\boldsymbol{D}'}{H_1} \cdot \boldsymbol{\zeta}_1 + \frac{\boldsymbol{C}'}{H_1} \cdot \boldsymbol{\zeta}_1 - \frac{\boldsymbol{D}'}{H_2} \cdot \boldsymbol{\zeta}_2 - H_1 \big[ \mathcal{F}^{-1}(\hat{\Phi})(-\boldsymbol{x}, -t) * \nabla \Pi_1 \big] +$$

$$+ \frac{\beta C_T B(x,y)}{H_1} \boldsymbol{U}' \cdot \boldsymbol{\zeta}_1 = 3\beta C_p B(x,y) \|\boldsymbol{U}_1\| \boldsymbol{U}_1 \qquad\qquad \text{in } \Omega \times (0, T],$$

$$-\frac{\partial \boldsymbol{\zeta}_2}{\partial t} - \boldsymbol{U}_2 \cdot \nabla \boldsymbol{\zeta}_2 + f_c \boldsymbol{J} \cdot \boldsymbol{\zeta}_2 - \nu_{t,2} \nabla^2 \boldsymbol{\zeta}_2 + \frac{\boldsymbol{D}'}{H_2} \cdot \boldsymbol{\zeta}_2 - \frac{\boldsymbol{D}'}{H_1} \cdot \boldsymbol{\zeta}_1 - H_2 \big[ \mathcal{F}^{-1}(\hat{\Phi})(-\boldsymbol{x}, -t) * \nabla \Pi_2 \big] = 0 \qquad \text{in } \Omega \times (0, T],$$

$$-\frac{\partial \Pi_1}{\partial t} - \boldsymbol{U}_1 \cdot \nabla \Pi_1 - \nabla \cdot \boldsymbol{\zeta}_1 - \nabla \cdot \boldsymbol{\zeta}_2 = 0 \qquad\qquad\qquad \text{in } \Omega \times (0, T],$$

$$-\frac{\partial \Pi_2}{\partial t} - \boldsymbol{U}_2 \cdot \nabla \Pi_2 - \nabla \cdot \boldsymbol{\zeta}_1 - \nabla \cdot \boldsymbol{\zeta}_2 = 0 \qquad\qquad\qquad \text{in } \Omega \times (0, T].$$

$$\tag{18}$$

Note that the minus sign in the argument of $\mathcal{F}^{-1}(\hat{\Phi})(-\boldsymbol{x}, -t)$ is not a result from classical integration by parts, but arrives from applying Fubini's theorem to the convolution term in Eq. 12 and Eq. 13 (see Appendix A2 for details). The adjoint variables are grouped in the vector $\boldsymbol{\psi}^* = [\boldsymbol{\zeta}_1, \boldsymbol{\zeta}_2, \Pi_1, \Pi_2]$ where $\boldsymbol{\zeta}_1 = (\zeta_1, \chi_1)$ and $\boldsymbol{\zeta}_2 = (\zeta_2, \chi_2)$ are the adjoint velocity perturbation fields in the wind-farm and upper layer, respectively, while $\Pi_1$ and $\Pi_2$ are the adjoint pressure perturbations. Using the solution of the adjoint equations, the gradient of the cost function is expressed as (see Appendix A3 for details)

$$\nabla \widetilde{\mathcal{J}} = \frac{\beta B(x,y)}{H_1} \left[ \|\boldsymbol{U}_1\| \boldsymbol{U}_1 \cdot \boldsymbol{\zeta}_1 - H_1 \|\boldsymbol{U}_1\| \frac{dC_p}{dC_T} \left( \|\boldsymbol{U}_1\|^2 + 3\boldsymbol{U}_1 \cdot \boldsymbol{u}_1 \right) + \boldsymbol{u}_1^\top \cdot \boldsymbol{U}' \cdot \boldsymbol{\zeta}_1 \right], \tag{19}$$

where $dC_p/dC_T$ is computed from Eq. 14. To compute the gradient $\nabla \widetilde{\mathcal{J}}$ we need to solve the forward and backward equations. Since the cost for solving the adjoint equations is roughly the same as for the forward equation, the computational cost for evaluating $\nabla \widetilde{\mathcal{J}}$ is proportional to the cost of solving twice the state equations. To verify the approach, we compare the adjoint gradient to a standard finite-difference approximation in Appendix A4.

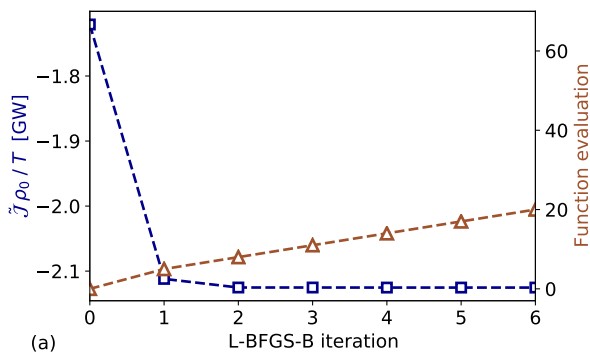 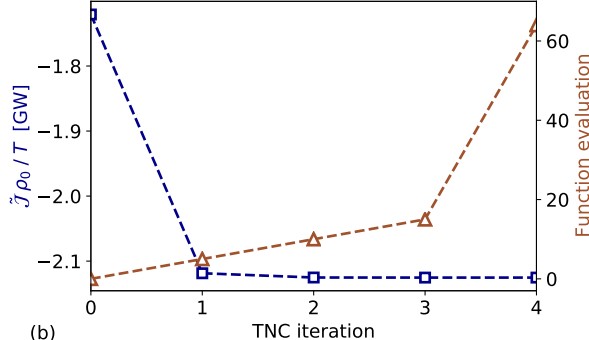

**Figure 1.** Convergence of the cost functional over the (a) L-BFGS-B and (b) TNC iteration. The squares and triangles denote the cost functional value and the number of function evaluations, respectively.

## 3 Numerical setup and case description

We define the model setup used to assess the potential of set-point optimization in mitigating self-induced gravity-wave effects in this section. We discuss the numerical setup in Section 3.1. Next, the selected wind-farm layout is presented in Section 3.2. Finally, the atmospheric state is discussed in Section 3.3.

### 3.1 Numerical setup

Both the forward and adjoint equations are discretized using a Fourier–Galerkin spectral method in space and time. The use of Fourier modes in time, automatically results in satisfying the periodicity conditions that we are aiming for in our optimization set-up. All terms in the equations are linear, besides the first-order term of the wind-farm drag force (and its adjoint). These terms contain products between temporarily and spatially dependent variables. To avoid expensive convolution sums, these products are computed in physical space. Full dealiasing is obtained by padding and truncation according to the 3/2 rule (Canuto et al., 1988). The use of Fourier modes in space forces periodic boundary conditions at the edges of the computational domain. Therefore, the domain has a sufficiently large dimension $D_x \times D_y = 1000 \times 400$ km$^2$, so that the perturbations die out before being recycled. The grid has $N_x \times N_y = 4000 \times 1600$ grid points which corresponds to a space resolution of $\Delta = 250$ m or $6.4 \times 10^6$ DOF per layer. Finally, different time horizon values are used spanning from $T = 10$ minutes to $T = 10$ hours with number of time steps ranging from $N_t = 12$ to $N_t = 120$. The discretized forward and backward equations form two systems of dimension $6N_xN_yN_t \times 6N_xN_yN_t$, which are solved using the LGMRES algorithm (Baker et al., 2005).

For the optimization, two different algorithms are compared in Fig. 1. The L-BFGS-B (limited-memory Broyden–Fletcher-Goldfarb–Shanno with box constraint) algorithm (Byrd et al., 1995) is an iterative quasi-Newton method. In the current application, the step length is evaluated with the inexact line search Wolfe condition (Wolfe, 1969). The truncated Newton method (TNC) computes the search direction by solving iteratively the Newton equation, applying the conjugate gradient method. This inner loop is stopped (truncated) as soon as a termination criterion is satisfied (Nocedal and Wright, 1999). In both cases, the

system matrix of the Newton equation consists of an approximate Hessian matrix, while the right-hand side needs gradient information to be computed, which are provided by the continuous adjoint method (see Appendix A for derivation and validation). Fig. 1 shows that the cost function decreases rapidly in the first two to three algorithm iterations, reaching a plateau afterwards. The use of a quasi-Newton method in combination with the limited complexity of our optimization model (for instance, the constraints are linearized equations) allow us to reach such a fast convergence. Moreover, the continuous adjoint method limits the number of function evaluations, since it is not necessary to evaluate $\widetilde{\mathcal{J}}(C_T + \alpha \delta C_T)$ for all directions $\delta C_T$ in the control space (at the expenses of solving an auxiliary set of equations). In particular, Fig. 1(a) displays that the cost functional is converged after six L-BFGS-B iterations. Apart from the first iteration, the line search method needs three 'function evaluations' before updating the cost functional. Hence, we need to solve 20 times the forward and backward equations for reaching convergence. On the other hand, Fig. 1(b) illustrates that the cost functional is mainly minimized within the first TNC iteration and convergence is reached after only four iterations. However, 63 function evaluations are needed. Hence, we will use the L-BFGS-B algorithm for solving the PDE-constrained optimization problem in the remainder of the article. To limit computational effort, a maximum of four L-BFGS-B iterations will be performed.

The solver (which is not parallelized) takes a couple of hours to solve the equations for a grid with resolution of 250 m ($6.4 \times 10^6$ DOF per layer). Since convergence is reached after approximately 20 function evaluations (which means that we solve state and adjoint equations 20 times), the optimizer takes a couple of days to compute an optimal thrust set-point distribution. However, after this work was performed, we have upgraded the forward solver which is now approximately 1000 times faster than our previous version. Optimization of the backward solver is planned for the future, and we expect that this will lead to an optimization algorithm that will only take several minutes for the same case.

## 3.2 Wind-farm layout

Allaerts and Meyers (2019) conducted a sensitivity study on the effects of wind-farm layout on gravity-wave induced power losses. They show that these power losses monotonically increase with the size of the farm. Also, they mention that the losses are maximum when the wind-farm ratio $L_y/L_x$ is close to $3/2$, while being negligible for very wide but short farm, and vice versa (i.e., negligible for $L_y/L_x \ll 1$ and $L_x/L_y \gg 1$). Since we are interested in optimal thrust coefficient distributions in presence of gravity waves, we have selected the "worst-case" wind-farm layout (i.e., a wind-farm width and length of $L_y = 30$ km and $L_x = 20$ km, respectively). We note that this was also the farm layout chosen by Allaerts et al. (2018) and Allaerts and Meyers (2019), which resembles in size the Belgian-Dutch wind-farm offshore cluster located in the North Sea, but simplified to a rectangular shaped wind-farm. Also, Smith (2010), Fitch et al. (2012) and Wu and Porté-Agel (2017) have used a farm with similar dimensions in their studies. The wind turbine relative spacings along the x- and y-direction are $s_x = s_y = 5.61$ (both non-dimensionalized with respect to the turbine rotor diameter $D$), so that the density of turbines in the farm is similar to the one of Allaerts and Meyers (2019) (i.e. leading to $\beta = 0.01$ in Eq. 8, setting both the wake efficiency $\eta_{\mathrm{w}}$ and $\gamma$ to 0.9 as in Allaerts and Meyers (2018)). Note that we do not define a specific layout or a number of turbines but we only fix the density of turbines in the farm. The turbine dimensions are based on a DTU 10-MW IEA wind turbine (Bortolotti et al., 2019) with rotor diameter $D = 198$ m and turbine hub height $z_h = 119$ m.

## 3.3 Background state variables

The governing equations are linearized around a constant background state. To determine this state, we need vertical profiles of potential temperature, velocity, shear stress and eddy viscosity plus the surface roughness $z_0$ and the friction velocity $u_*$. We describe the techniques used in determining these profiles in the remainder of this section. Similar to Allaerts and Meyers (2019), we select two atmospheric states for initial testing of the optimizer, corresponding to sub- and supercritical flow conditions.

We choose a temperature profile that corresponds to a conventionally neutral ABL. The potential temperature in the neutral ABL is fixed to $\theta_0 = 288.15$ K. A capping inversion strength $\Delta\theta$ of 5.54 K and 3.7 K is used, which lead to a sub- and supercritical flow, respectively (see below). Finally, a free atmosphere lapse rate $\Gamma = 1$ K/km is chosen. The lapse rate also defines the Brunt-Väisälä frequency $N$.

The velocity and stress profiles within the ABL are obtained following the approach of Nieuwstadt (1983). The non-dimensional surface roughness length $\overline{z}_0 = z_0/H$ and the non-dimensional boundary-layer height $h_* = Hf_c/u_*$ are the input parameters of Nieuwstadt's model, where $f_c$ is the Coriolis frequency and $H = H_1 + H_2$ is the ABL height. The wind-farm layer height is assumed to be twice the turbine hub height, so $H_1 = 2z_h$. The ABL height is fixed to $H = 1000$ m and the friction velocity is set to $u_* = 0.6$ m/s. Finally, a surface roughness of $z_0 = 10^{-1}$ m is adopted. Using $h_* = 0.166$ and $\overline{z}_0 = 10^{-4}$ as input values for the Nieuwstadt's model, we derive the velocity $\boldsymbol{U}_1$, $\boldsymbol{U}_2$, the eddy viscosity $\nu_{t,1}$, $\nu_{t,2}$ for the wind-farm and upper layer and the friction coefficients $C$ and $D$ (used for computing the matrices $\boldsymbol{C}'$ and $\boldsymbol{D}'$, see Allaerts and Meyers (2019)). Besides the friction coefficients $C$ and $D$ which are given at $z = 0$ and $z = H_1$, all other physical quantities are depth-averaged over the height $H_1$ and $H_2$. Finally, the wind profile is oriented such that the wind in the wind-farm layer is always directed along the x-axis (i.e, $V_1 = 0$ m/s).

The pressure gradient strengths induced by inversion and internal gravity waves are dependent upon the Froude number $F_r = U_B/\sqrt{g'H}$ and a non-dimensional group $P_N = U_B^2/NH\|\boldsymbol{U}_g\|$, respectively (Smith, 2010; Allaerts and Meyers, 2019), where the velocity scale $U_B$ is defined as

$$U_B = \left( \frac{H_1}{H} \frac{1}{U_1^2} + \frac{H_2}{H} \frac{1}{U_2^2} \right)^{\frac{1}{2}}. \tag{20}$$

The chosen background state defines a Froude number of 0.9 for $\Delta\theta = 5.54$ K, which implies subcritical flow conditions ($F_r < 1$), and a Froude number of 1.1 for $\Delta\theta = 3.7$ K, which leads to supercritical flow conditions ($F_r > 1$). Further, $P_N$ expresses the impact of internal waves in the troposphere which increases when $P_N$ decreases. The background state defined in Table 1 leads to $P_N = 1.92$. The numerical setup, wind-farm layout and background state variables are summarized in Table 1.

**Table 1.** Numerical setup, wind-farm layout and atmospheric state used in this manuscript.

| Numerical setup | |
| --- | --- |
| Domain size | $D_x \times D_y = 1000 \times 400 \text{ km}^2$ |
| Grid size | $N_x \times N_y = 4000 \times 1600$ |
| Grid resolution | $\Delta = 250 \text{ m}$ |
| Time horizon | Span from $T = 10$ min to $T = 10$ h |
| Time step | Span from $N_t = 12$ to $N_t = 120$ |
| Discretization technique | Fourier–Galerkin |
| Equation solver | LGMRES |
| Optimization method | L-BFGS-B |
| L-BFGS-B iterations | $N_{it} = 4$ |
| **Wind-farm layout** | |
| Wind-farm length | $L_x = 20 \text{ km}$ |
| Wind-farm width | $L_y = 30 \text{ km}$ |
| Turbine hub height | $z_h = 119 \text{ m}$ |
| Turbine rotor diameter | $D = 198 \text{ m}$ |
| Rated wind speed | $U_r = 11 \text{ m/s}$ |
| Relative turbine spacing | $s_x = s_y = 5.61$ |
| Wake efficiency | $\eta_w = 0.9$ |
| Velocity shape factor | $\gamma = 0.9$ |
| **Atmospheric state** | |
| ABL potential temperature | $\theta_0 = 288.15 \text{ K}$ |
| Capping inversion strength | $\Delta\theta = 5.54 \text{ K} \rightarrow F_r = 0.9$ |
| | $\Delta\theta = 3.70 \text{ K} \rightarrow F_r = 1.1$ |
| Free atmosphere lapse rate | $\Gamma = 1 \text{ K/km}$ |
| Surface roughness | $z_0 = 10^{-1} \text{ m}$ |
| Coriolis frequency | $f_c = 10^{-4} \text{ 1/s}$ |
| Friction velocity | $u_* = 0.6 \text{ m/s}$ |
| Boundary layer height | $H = 1000 \text{ m}$ |
| Friction coefficients | $C = 3.76 \times 10^{-3}$ |
| | $D = 1.51 \times 10^{-1}$ |

## 4 Results and discussion

We discuss the results of the optimization problem in the current section. Firstly, the optimal thrust-coefficient distributions and relative power gains are illustrated for two specific flow conditions in Section 4.1. Thereafter, the sensitivity of the power gain to the atmospheric state is studied in Section 4.2.

## 4.1 Optimal thrust-coefficient distributions

The optimization model described in Section 2.2 is time and space dependent. Hence, the model is capable of finding a time-periodic optimal thrust-coefficient distribution over the wind-farm area in a fixed time interval $[0, T]$. However, all optimal thrust set-point distributions found for the different combinations of time horizons and time steps reported in Table 1, are constant in time. We have verified this using a range of steady and unsteady starting conditions for $C_T$ in the algorithm, but did not find any unsteady optimum. We believe that this is due to the use of steady-state inflow conditions, meaning that we neglect meso-scale temporal variations in the velocity field (these could lead to time-dependent optimal control signals, but are not included in the current work). Since we do not observe any unsteady behaviour in our optimal solutions, we show only steady-state results in the remainder of the manuscript, and conclude for the time being that unsteady time-periodic excitation is less effective than a stationary spatially optimal distribution in this context.

We also note that our findings are in contrast with recent works of Goit and Meyers (2015), Munters and Meyers (2018) and Frederik et al. (2020), in which the authors illustrated the benefits of dynamic induction control over yaw and static induction control. However, the characteristic time scale of gravity-wave effects is estimated to be approximately 1 h (Gill 1982, Allaerts and Meyers 2019) which is an order of magnitude above the typical time scale of wake convection between turbines, and turbulent mixing in turbine wakes (this also justifies the larger sampling time used). Hence, while unsteadiness of the thrust coefficient (with a typical time scale of 50 seconds for large scale turbines) can lead to improved wake mixing (Goit and Meyers, 2015; Munters and Meyers, 2018; Frederik et al., 2020), it has no impact on phenomena that occur at larger time scales, such as wind-farm induced gravity waves.

The steady-state optimal thrust-coefficient distributions obtained in sub- and supercritical conditions are analyzed in the remainder of this section. To improve the understanding of such distributions, gravity-wave induced flow patterns obtained with $C_T^O(x, y)$ are compared with a reference case. The setup of the reference model is the one reported in Table 1 but instead a uniform thrust set-point distribution over the wind-farm area is used, with $C_T^R(x, y) = C_T^{Betz} = 8/9$.

Figure 2 illustrates a planform view of the perturbation flow patterns obtained with $F_r = 0.9$ (top row) and $F_r = 1.1$ (bottom row) using the reference model setup. The farm extracts energy from the flow, causing a momentum sink in the wind-farm layer. Due to the continuity constraint, an upward flow displacement above the wind-farm area takes place which causes the boundary layer height to increase. Fig. 2(a) shows for the subcritical case, that an inversion-layer vertical displacement of about 65 meter takes place at the wind-farm entrance region. A second peak of lower magnitude is located in the downwind region. On the other hand, for the supercritical case, Fig. 2(d) displays a similar maximum value of $\eta_t$ attained close to the wind-farm center. In both cases, the inversion-layer vertical displacement decreases in the wind-farm exit region and assumes a wavy behaviour in the wind-farm wake. The vertical displacement of air parcels triggers inversion waves on the 2D inversion layer surface and internal waves in the free atmosphere (3D waves). These waves induce pressure gradients, as visible in Fig. 2(b,e), where a region of high pressure builds up in correspondence with high $\eta_t$ values, leading to flow blockage. However, Fig. 2(b) shows a stronger adverse pressure gradient in the wind-farm induction region than the one in Fig. 2(e). In fact, inversion waves travel upstream in subcritical conditions, which leads to more slow-down in the induction region. In both sub- and supercritical

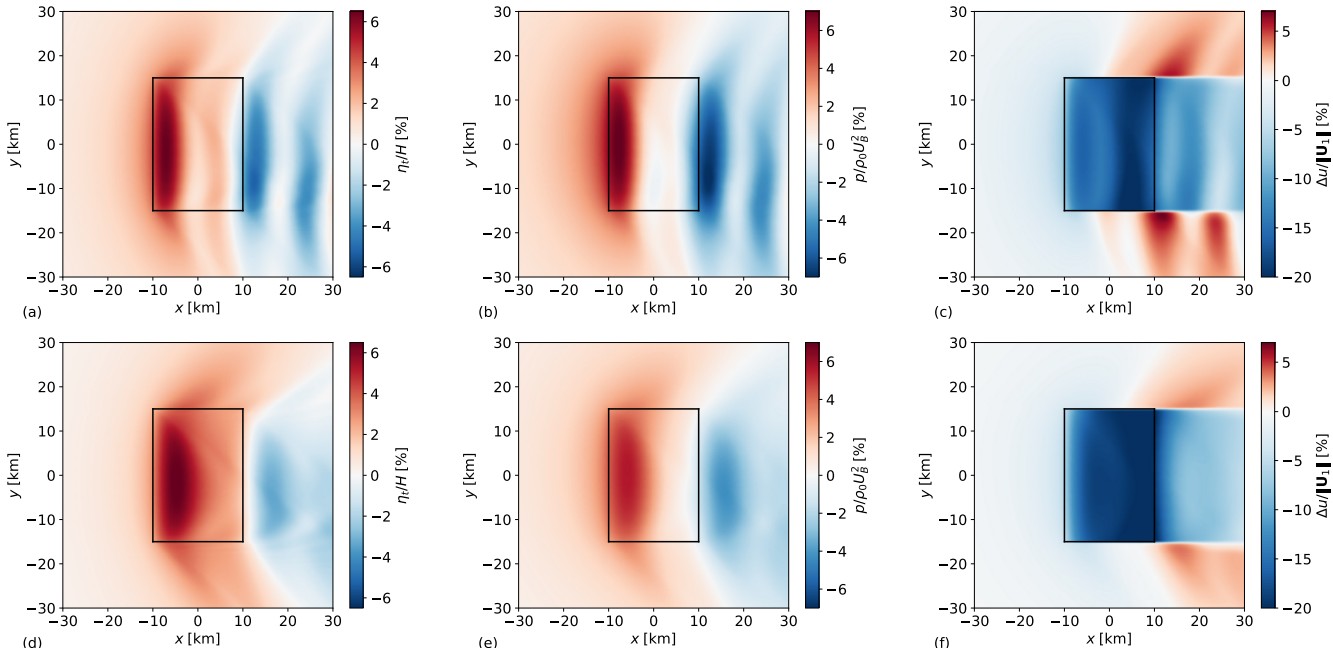

**Figure 2.** Planform view of inversion-layer displacement (a,d), pressure perturbation (b,e) and relative velocity reduction (c,f) in the wind-farm layer in subcritical (top row, $F_r = 0.9$) and supercritical (bottom row, $F_r = 1.1$) flow conditions. The black rectangle indicates the wind-farm region.

case, favourable pressure gradients reduce the velocity deficits in the bulk of the farm. Finally, Fig. 2(c,f) illustrate relative velocity reductions in the wind-farm layer. The stronger inversion strength found in the subcritical flow case transforms the inversion layer in a quasi-rigid lid, which limits vertical displacements. The lower streamlines divergence over the wind-farm area implies lower velocity reductions. Moreover, the favourable pressure gradient is stronger when $F_r = 0.9$, allowing for lower velocity deficits within the wind-farm area. This explains the higher velocity reduction (up to 20%) seen in Fig. 2(f). Such a strong response could be on the limit of our small amplitude assumption. The planform view of pressure and velocity perturbations in the wind-farm and upper layer in subcritical flow conditions are also illustrated on a wider domain in Appendix A (see Fig. A1).

The goal of our study is to find an optimal set-point distribution which reduces the velocity perturbations displayed in Fig. 2(c,f). While maximizing the flow wind speed through the farm, we also maximize the wind-farm energy extraction. To this end, we solve the optimization problem discussed in Section 2.2. The inputs of the optimization model are the wind-farm layout and the atmospheric conditions, which are detailed in Table 1. Moreover, an initial thrust-coefficient distribution needs to be specified. We have verified that for many different initial conditions the algorithm converges always to the same optimal solution. Therefore, a random initial thrust set-point distribution is chosen. The optimal configurations obtained for different Froude numbers are illustrated in Fig. 3. We find that the optimal thrust-coefficient distributions are non-uniform in space and

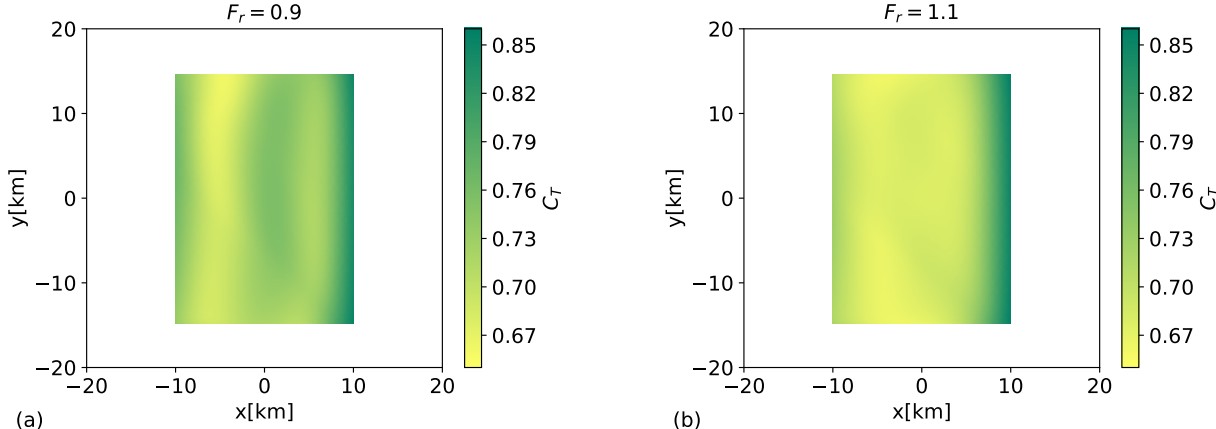

**Figure 3.** Planform view of (a) optimal thrust-coefficient distribution in subcritical ($F_r = 0.9$) and (b) supercritical flow conditions ($F_r = 1.1$). The length and width of the wind farm are 20 and 30 km, respectively.

assume different spatial distributions according to the atmospheric state. In particular, when the flow is subcritical the optimal thrust set-point distribution assumes a sinusoidal behaviour in the streamwise direction while it becomes a U-shaped curve when the flow is supercritical. In both cases, $C_T^O$ is almost invariant along the spanwise direction.

We denote with $\mathcal{P}^R = \widetilde{\mathcal{J}}^R/T$ and $\mathcal{P}^O = \widetilde{\mathcal{J}}^O/T$ the power extracted using $C_T = C_T^R = 8/9$ and $C_T = C_T^O$, respectively. Further, we define

$$\mathcal{G} = \frac{\mathcal{P}^O - \mathcal{P}^R}{\mathcal{P}^R} \tag{21}$$

where $\mathcal{G}$ denotes the relative power gain obtained using an optimal thrust-coefficient distribution instead of the reference one. Note that the optimal distributions are steady-state, therefore the power gain definition is not dependent on the choice of the

time horizon $T$. The power gains attained in sub- and supercritical flow conditions are 5.3% and 7%, respectively. Clearly, power gains are also strongly dependent on the atmospheric conditions. Therefore, a sensitivity study is carried out in Section 4.2. To asses the benefit of an optimal non-uniform distribution over an optimal uniform one, we have applied the optimization framework developed in section 2.2 assuming a spatially invariant $C_T$. Results are discussed in Appendix B.

The optimal set-point distributions displayed in Fig. 3 are related to the vertical displacement of the inversion layer over the

wind-farm area. Fig. 4 shows streamwise profiles of $\eta_t$ and $C_T^O$ through the center of the farm for $F_r = 0.9$ and $F_r = 1.1$. To reduce gravity-wave excitation, $C_T^O$ is seen to be inversely related with $\eta_t$. In fact, Fig. 4(a) shows that the streamwise profile of $\eta_t$ has a sinusoidal behaviour. Hence, the optimal set-point distribution is sinusoidal as well, explaining the pattern displayed in Fig. 3(a). On the other hand, $\eta_t$ assumes a U-shaped profile through the wind farm in supercritical conditions (see Fig. 4(b)), a profile that is also found in $C_T^O$ (see Fig. 3(b)). Moreover, Fig. 2(a,d) show that the gradient of $\eta_t$ along the spanwise direction

is much smaller than the one along the streamwise direction, explaining the almost constant thrust set-point distributions along the y-direction. Fig. 4 also shows that $\eta_{t,\max}^O < \eta_{t,\max}^R$ in both sub- and supercritical conditions, meaning that the optimal

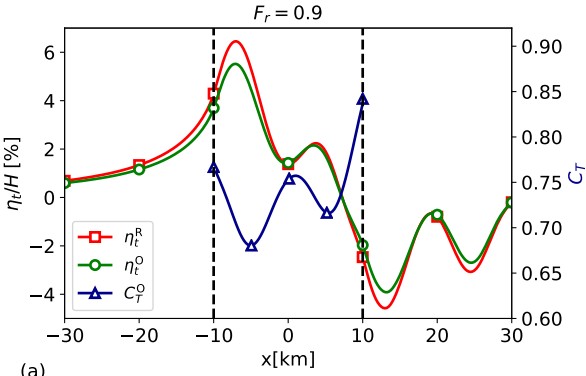
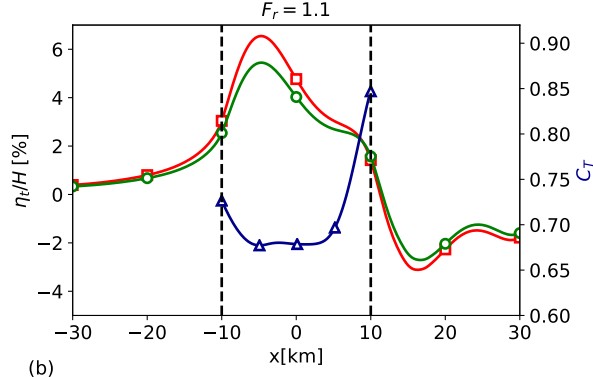

(a)  (b)

**Figure 4.** Streamwise profiles of optimal thrust set-point distribution ($C_T^{\mathrm{O}}$), reference ($\eta_t^{\mathrm{R}}$) and optimal ($\eta_t^{\mathrm{O}}$) inversion-layer displacement in (a) subcritical ($F_r = 0.9$) and (b) supercritical flow conditions ($F_r = 1.1$). The wind-farm region is marked by vertical dashed lines, and the profiles have been obtained through the centre of the farm ($y = 0$).

thrust set-point distribution decreases the upward flow displacement over the wind-farm area. The maximum inversion-layer displacement is located at the entrance region of the farm. If we compare $\eta_t^{\mathrm{R}}$ and $\eta_t^{\mathrm{O}}$ in this region, a displacement reduction of 14.5% and 16.8% is attained with the optimal configuration for the sub- and supercritical case, respectively.

A lower vertical displacement of the inversion layer reduces gravity-wave excitation, therefore we also expect a lower strength of the adverse pressure gradient at the entrance of the farm compared to the one obtained with $C_T^{\mathrm{R}}$. Fig. 5(a,c) confirm this hypothesis, showing streamwise profiles of pressure perturbations $p^{\mathrm{R}}$ and $p^{\mathrm{O}}$ through the center of the farm for $F_r = 0.9$ and $F_r = 1.1$. The pressure peak is located at the entrance of the farm and a pressure peak reduction of 14.3% and 16.2% is attained with the optimal configuration for the sub- and supercritical case, respectively. Fig. 5(b,d) show streamwise profiles

of velocity perturbations $u_1^{\mathrm{R}}$ and $u_1^{\mathrm{O}}$ through the center of the farm for $F_r = 0.9$ and $F_r = 1.1$. The lower adverse pressure gradient strength attained with the optimal configuration allows for a lower velocity perturbation $u_1$ in the induction region with respect to the reference case. Moreover, the optimal configuration also reduces the streamline divergence, accounting for higher flow wind speeds through the farm. Consequently, a velocity perturbation reduction of 13.4% and 15.5% is attained for the sub- and supercritical case, which explains the higher power gain obtained for $F_r = 1.1$. The relative change in percentage

between optimal and reference maximum flow perturbation values are summarized in Table 2.

    The optimal thrust-coefficient distributions and power gains discussed in this section are obtained with data listed in Table 1. However, the atmospheric state changes in real case scenarios and we have seen that the optimal configuration strongly depends upon the atmospheric parameters. Therefore, the sensitivity of the power gain to the atmospheric state is performed in the next section.

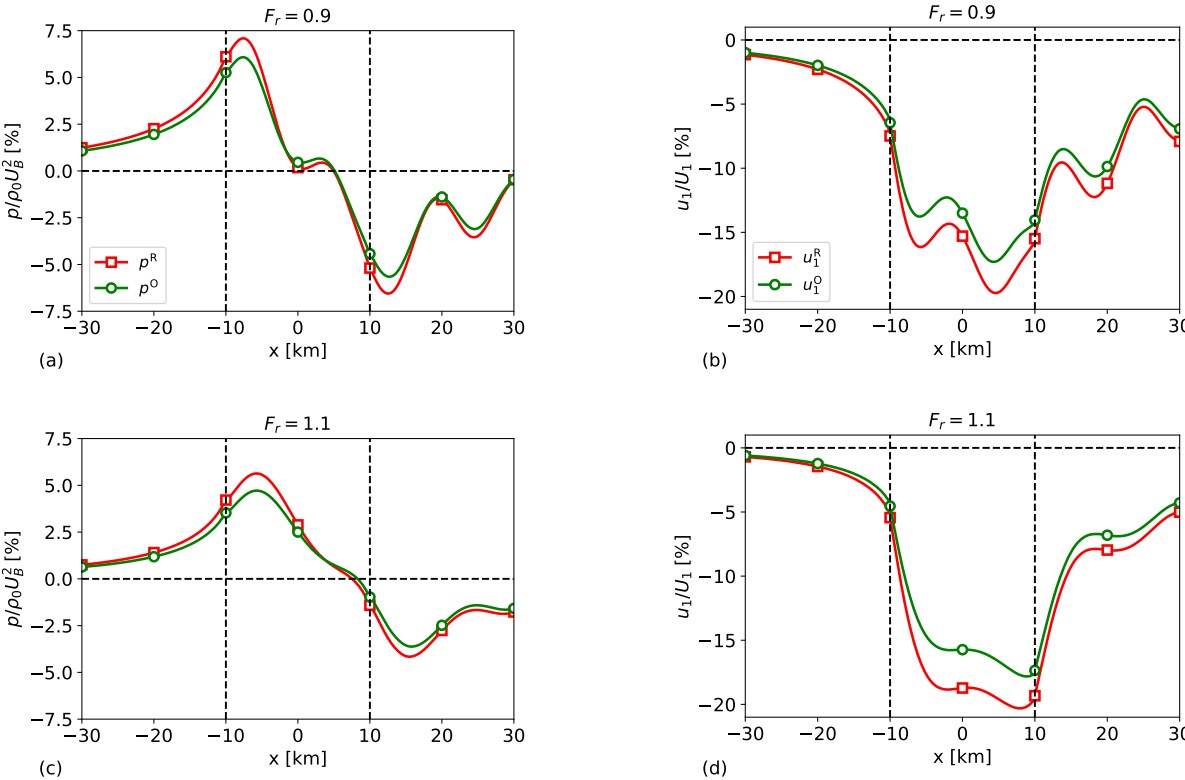

**Figure 5.** Streamwise profiles of (a,c) reference ($p^R$) and optimal ($p^O$) pressure perturbation and (b,d) reference ($u_1^R$) and optimal ($u_1^O$) velocity perturbation in subcritical (top row, $F_r = 0.9$) and supercritical (bottom row, $F_r = 1.1$) flow conditions. The wind-farm region is marked by vertical dashed lines, and the profiles have been obtained through the centre of the farm ($y = 0$).

**Table 2.** Relative change in percentage between optimal and reference maximum flow perturbation values. Power gains are also included.

|  | $\mathbf{F_r = 0.9}$ | $\mathbf{F_r = 1.1}$ |
| --- | --- | --- |
| Maximum inversion-layer displacement | -14.5% | -16.8% |
| Maximum pressure perturbation | -14.3% | -16.2% |
| Maximum velocity perturbation | -13.4% | -15.5% |
| Power gain | 5.3% | 7.0% |

## 4.2 Sensitivity study

Allaerts and Meyers (2019) pointed out that gravity-wave induced power loss is significant only for certain atmospheric states. Since our aim is to recover its power loss, we also expect the power gain to be sensitive to the atmospheric conditions. We note

that gravity-wave patterns are also sensitive to the wind-farm layout. However, a sensitivity study over the wind-farm layout is beyond the scope of the article.

The nondimensionalization of the three-layer model equations with respect to the boundary layer height $H$ and the friction velocity $u_*$ highlights four non-dimensional groups that govern the atmospheric state, which are:

- The non-dimensional boundary layer height $h_* = H f_c / u_*$. Values of $h_* \approx 0.1$ denote shallow boundary layers typically found over sea, while $h_* \approx 0.35$ rather relates to a deep land-based boundary layer. We vary $h_*$ between 0.16 and 0.4;

- The non-dimensional surface roughness length $\overline{z}_0 = z_0 / H$. This number varies on several order of magnitude according to the sea state or land surface. We vary $\log_{10}(\overline{z}_0)$ between $-4.2$ and $-2.8$ in the current study;

- The non-dimensional Brunt-Väisälä frequency $N/f_c$. The Brunt-Väisälä frequency is an important parameter in gravity-wave theory which expresses the highest possible frequency for internal gravity waves (Gill, 1982). Typical values of free atmosphere lapse rate $\Gamma$ range between 1 and 10 K/km. Low and high $\Gamma$ values are associated with weakly and strongly stratified atmospheres, respectively. We vary $\Gamma$ between 0.03 and 12 K/km corresponding to $10 \leq N/f_c \leq 200$;

- The inversion parameter $g'H/Au_*^2$. According to Csanady (1974), the height of the inversion layer is determined by a balance of surface stress and buoyancy. Equilibrium conditions are reached when $g'H/Au_*^2 \approx 1$, with $A = 500$ being an empirical constant. We vary the inversion parameter between 0.5 and 1.5.

Allaerts and Meyers (2019) conducted a similar sensitivity study on the gravity-wave induced power loss on a wider range of non-dimensional numbers. However, since we are optimizing turbine thrust set points, we need to ensure that $U_1/U_r < 1$

($U_r = 11$ m/s is the rated wind speed of the DTU 10 MW IEA wind turbine), otherwise turbines would operate in above-rated wind speed regime and it would not make any sense to optimize their power production. The choice of the four ranges for the non-dimensional groups discussed above ensures that $U_1/U_r \leq 0.9$ for all atmospheric states.

    Using the atmospheric state reported in Table 1, the non-dimensional numbers assume values $h_* = 0.166$, $\overline{z}_0 = 10^{-4}$ and $N/f_c = 58$. The inversion parameter is equal to 1.046 and 0.691 in the sub- and supercritical case, respectively. The optimal

thrust-coefficient distributions discussed in Section 4.1 were obtained using these dimensionless group values. The sensitivity of the power gain to atmospheric conditions is performed by varying $h_*$, $\overline{z}_0$ and $N/f_c$ against the inversion parameter $g'H/Au_*^2$, similarly to Allaerts and Meyers (2019). The numerical setup is the one detailed in Table 1. However, we use a grid cell size which is four times bigger ($\Delta x \times \Delta y = 1000 \times 1000$ m$^2$), meaning that we use $4 \times 10^5$ cells instead of $6.4 \times 10^6$, so that the necessary computational resources remain reasonable. To assess the validity of this choice, we performed a grid sensitivity

study in Appendix C showing that the power gain value changes of about 1% when the number of grid cells is increased of one order of magnitude (see Fig. C1). The high computational efficiency of the three-layer model allowed us to perform a sensitivity study of the optimization results over 1960 different atmospheric conditions (thus effectively running an optimization problem for every atmospheric state). Since the wind-farm layout impact on power gains is beyond the scope of our study, we impose the wind direction to be along the $x$-axis in the wind-farm layer in all simulations ($V_1 = 0$ m/s).

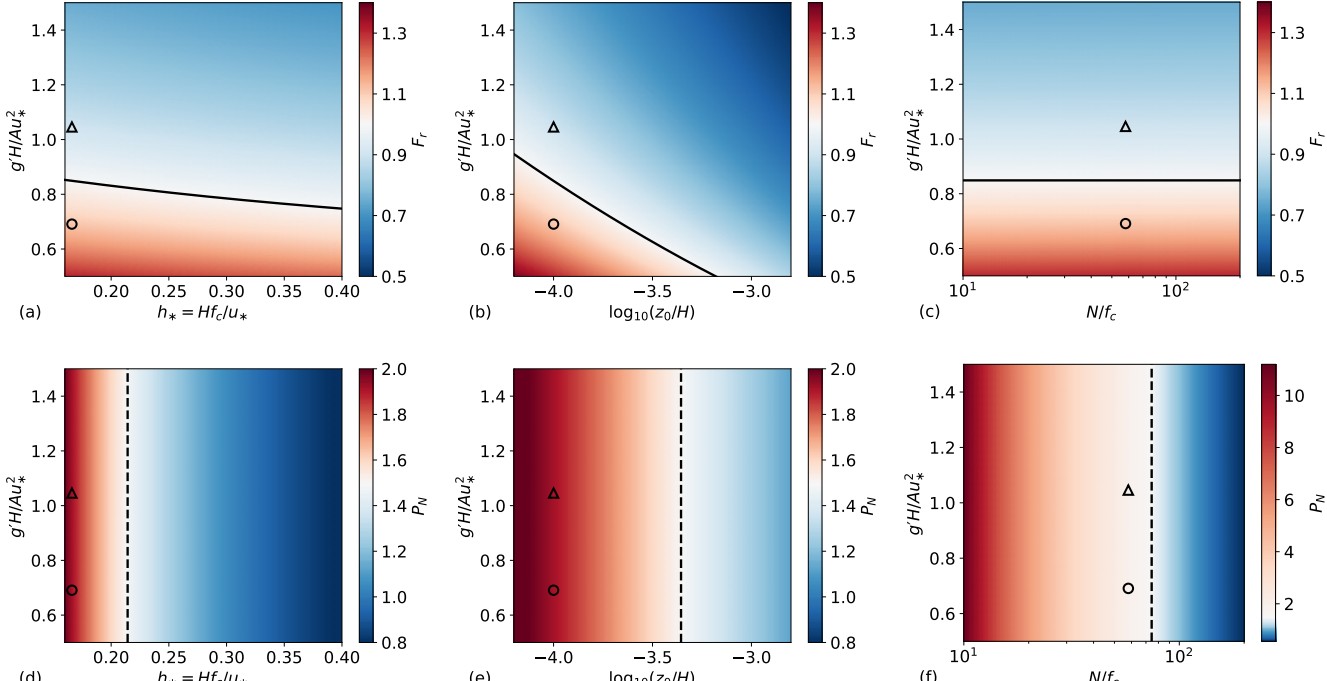

**Figure 6.** Sensitivity of (a-c) Froude number $F_r = U_B/\sqrt{g'H}$ and (d-f) $P_N = U_B^2/NH\|\boldsymbol{U}_g\|$ to atmospheric conditions. (a,d) Non-dimensional boundary layer height $h_*$ (with $\overline{z}_0 = 10^{-4}$ and $N/f_c = 58$), (b,d) logarithm of the non-dimensional surface roughness length $\overline{z}_0$ (with $h_* = 0.166$ and $N/f_c = 58$) and (c,f) ratio of Brunt-Väisälä frequency to Coriolis parameter $N/f_c$ (with $h_* = 0.166$ and $\overline{z}_0 = 10^{-4}$) against the inversion parameter $g'H/Au_*^2$. The black solid lines in (a-c) corresponds to critical flow conditions ($F_r = 1$) while the dashed black ones in (d-f) corresponds to flow conditions of $P_N = 1.5$. The markers $\triangle$ and $\circ$ represent the sub- and supercritical flow case studied in Section 4.1, respectively. Note that Fig. (f) use a different scale than Fig. (d,e).

To better understand the power gain sensitivity to atmospheric conditions, we examine how the non-dimensional parameters $F_r$ and $P_N$ impact the flow fields. The pressure gradients induced by inversion waves scale with $g'$, therefore high inversion strengths correspond to strong inversion-wave feedback and low Froude number values. These two-dimensional waves are non-dispersive with phase speed $\sqrt{g'H}$ (Sutherland, 2010). Therefore, $F_r$ also represents the ratio of the bulk wind speed within the ABL to the velocity of the inversion waves. If $F_r < 1$ (subcritical flow) the two-dimensional waves can affect the

upstream flow, while they can travel only downstream if $F_r > 1$ (supercritical flow). The flow is said to be critical when $F_r = 1$. On the other hand, internal-wave induced pressure gradients are governed by the second non-dimensional group $P_N$. Strong internal-wave feedback correspond to low $P_N$ values. In fact, strongly stratified atmospheres imply high $N$ values, meaning that they account for higher internal-wave oscillation frequencies and phase speed (Sutherland, 2010).

     Two different flow regimes can be identified:

– Regime 1: low $P_N$. The strongly stratified free atmosphere limits vertical displacement of air parcels, hence reduced streamline divergence over the wind-farm area is observed. This results in low velocity reductions and $\eta_t$ values. Moreover, the flow fields are $F_r$-independent in these atmospheric states (Smith, 2010).

     – Regime 2: high $P_N$. The inversion-layer strength determines the flow fields properties since the influence of internal waves is negligible. The weakly stratified atmosphere makes the ABL to behave like an idealized shallow-water system

for $F_r \simeq 1$ (choking effect (Smith, 2010)). Moreover, the perturbations magnitude are strongly dependent upon the Froude number.

Smith (2010) and Allaerts and Meyers (2019) defined a third regime where $N = 0$ and $g' = 0$, which would correspond to $F_r, P_N \to \infty$ or to a purely neutral atmosphere. Gravity waves are not excited in this particular flow condition and only drag forces and frictional effects play a role in the flow behaviour. Since we are interested in finding optimal thrust set-point

distributions which allow to recover gravity-wave induced power loss, we did not investigate this regime in the current study.

     Fig. 6(a-c) illustrate the sensitivity of $F_r$ to changes in $h_*$, $\overline{z}_0$ and $N/f_c$ against the inversion parameter. In all cases, the Froude number ranges from approximatively 0.5 to 1.4. The black line denotes critical flow conditions. Lines of constant Froude number run parallel to this line, meaning that $F_r$ is invariant and quasi-invariant to $N/f_c$ and $h_*$, respectively. On the other hand, changes in $\overline{z}_0$ have a strong impact on the wind profile convexity and therefore on $F_r$. The sensitivity of $P_N$ to

475 the atmospheric state is displayed in Fig. 6(d-f). $P_N$ is not dependent on the inversion parameter. Hence, lines of constant $P_N$ values are vertical and parallel to the dashed black line, which denotes atmospheric conditions for which $P_N = 1.5$. This line divides the domain in regions where the internal-wave effects are important (regime 1, right side) or limited (regime 2, left side). However, internal waves still play a crucial role in softening the flow perturbations magnitude when $P_N$ values are only slightly greater than 1, as in Fig. 6(d,e). On the other hand, very high $P_N$ numbers ($P_N > 10$) are attained in weakly stratified

conditions (see Fig. 6(f)). We will use the above mentioned regimes classification as a proxy for the interpretation of the power gain sensitivity patterns (note that the term high and low in the regimes characterization are referred to the maximum and minimum $F_r$ and $P_N$ values found over the sensitivity domain).

     Fig. 7(a-c) and Fig. 7(d-f) illustrate the sensitivity of the optimal inversion-layer vertical-displacement reduction $\mathcal{G}_\eta$, and power gain $\mathcal{G}$ to changes in $h_*$, $\overline{z}_0$ and $N/f_c$, against the inversion parameter. The displacement reduction is defined as

$$\mathcal{G}_\eta = \left| \frac{\eta_{t,\max}^{\mathrm{O}} - \eta_{t,\max}^{\mathrm{R}}}{\eta_{t,\max}^{\mathrm{R}}} \right| \tag{22}$$

where $\eta_{t,\max}^{\mathrm{O}}$ and $\eta_{t,\max}^{\mathrm{R}}$ denote the maximum inversion-layer displacement attained with the optimal and reference model configuration, respectively. As we discussed in Section 4.1, the lowering of the inversion-layer vertical displacement reduces the strength of the adverse pressure gradient, increasing the flow wind speed and consequently the wind-farm power output. Fig. 7 confirms this statement. In fact, regions of high vertical displacement reduction strictly correspond to regions of high

power gain.

     Allaerts and Meyers (2017, 2018, 2019) found that for low ABL heights, gravity waves induce strong pressure gradients and play an important role in the distribution of the kinetic energy within the farm. Indeed, the large geostrophic wind angle found

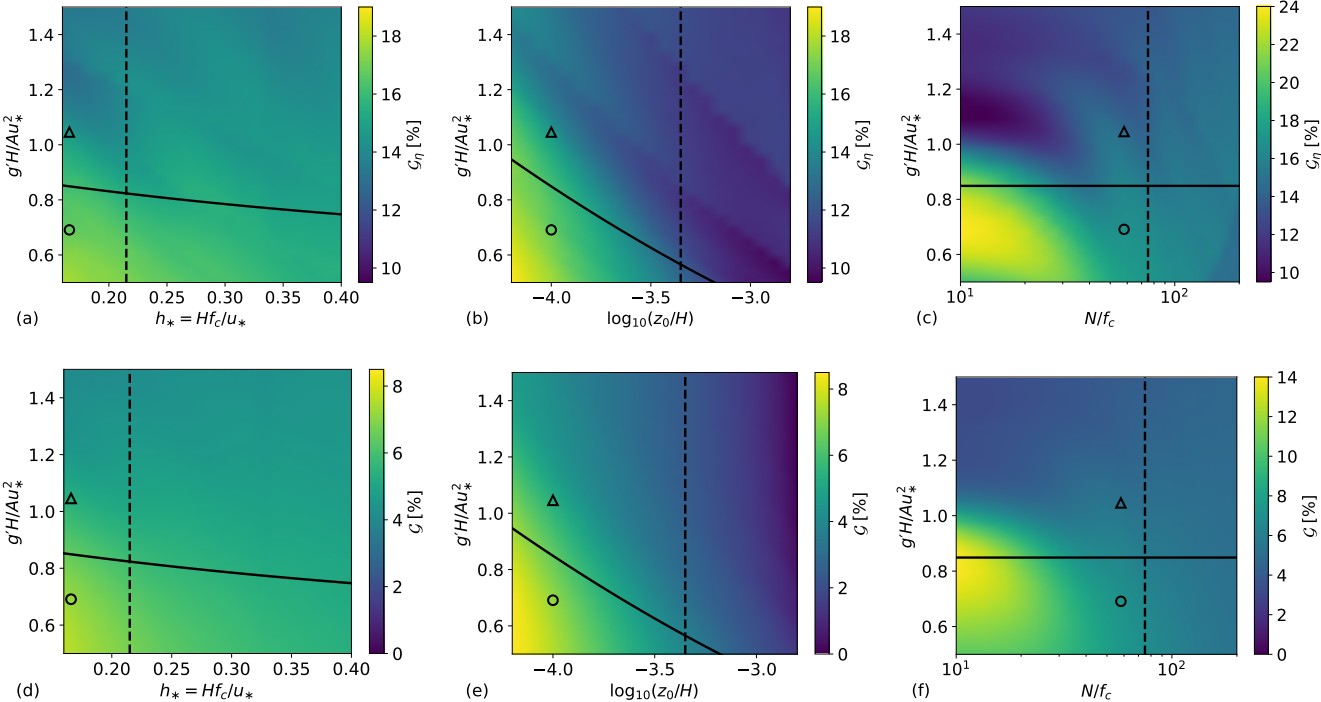

**Figure 7.** Sensitivity of (a-c) inversion-layer vertical-displacement reduction $\mathcal{G}_\eta$ and (d-f) power gain $\mathcal{G}$ to atmospheric conditions. (a,d) Non-dimensional boundary layer height $h_*$ (with $\overline{z}_0 = 10^{-4}$ and $N/f_c = 58$), (b,d) logarithm of the non-dimensional surface roughness length $\overline{z}_0$ (with $h_* = 0.166$ and $N/f_c = 58$) and (c,f) ratio of Brunt-Väisälä frequency to Coriolis parameter $N/f_c$ (with $h_* = 0.166$ and $\overline{z}_0 = 10^{-4}$) against the inversion parameter $g'H/Au_*^2$. The black solid line corresponds to critical flow conditions ($F_r = 1$) while the black dashed line corresponds to flow conditions of $P_N = 1.5$. The markers $\triangle$ and $\circ$ represent the sub- and supercritical flow case studied in Section 4.1, respectively. Note that Figs. (c) and (f) use a different scale than Figs. (a,b) and (d,e).

in shallow boundary layers redirects the favourable pressure gradient seen over the wind-farm area of 90 degrees for $h_* \to 0$, decreasing the dispersive impact of internal gravity waves. Fig. 7(d) displays that the maximum power gain is indeed attained

for $h_* = 0.17$ (i.e., for shallow boundary layer) in supercritical flow conditions, with gains of about 7.5% in correspondence to a displacement reduction of 17.5%. A similar pattern is seen in Fig. 7(e), where a maximum power gain of 8.4% is attained again in supercritical conditions for $\log_{10}(\overline{z}_0) = -4.2$, in correspondence to a displacement reduction of 19%. Both $\mathcal{G}$ and $\mathcal{G}_\eta$ show higher sensitivity to changes in $\overline{z}_0$ than in $h_*$, decreasing rapidly for increasing value of surface roughness. Interestingly, power gains are close to zero in case of high $\overline{z}_0$ values. This is due to the additional frictional drag which dissipates perturbation

energy, limiting gravity-wave excitation and consequently the potential of our optimization.

     The sensitivity of $\mathcal{G}$ and $\mathcal{G}_\eta$ to changes in free atmosphere stability are shown in Fig. 7(c,f). The high $P_N$ sensitivity to changes in $N$ (from $P_N \approx 11$ to $P_N \approx 0.5$ for increasing values of $N/f_c$) accounts for a clear distinction between regime 1 and regime 2. The former shows power gains of about 5% while the latter attains gains of 14% in correspondence to inversion

displacement reductions of 24%. Fig. 7(f) illustrates that the power gain peak is obtained in critical flow conditions ($F_r = 1$), differently from the previous cases. The very high $P_N$ values (hence, the limited presence of internal waves) attained in correspondence to $F_r = 1$ allow for the choking effect to take place (Smith, 2010; Allaerts and Meyers, 2019). Very large flow perturbations are triggered in these atmospheric conditions, leaving greater potential for power recovery. The choking effect is not visible in Fig. 7(d,e), since there $P_N \approx 2$ when $F_r = 1$ (the flow perturbations are softened by internal waves).

Overall, higher inversion-layer displacement reductions and power gains are attained in critical and supercritical flow conditions for high $P_N$ values (regime 2), that is for low $h_*$, $\overline{z}_0$ and $N/f_c$. This is not surprising due to the strong impact that gravity waves have on farm's performance in such conditions (see Section 4.1 or Smith (2010) and Allaerts and Meyers (2019)). Moreover, we observe strong gradients of $\mathcal{G}$ and $\mathcal{G}_\eta$ along contours of $P_N$ in regime 2, and weak gradients in regime 1. This suggests that the flow properties are $F_r$-independent for low $P_N$ values, confirming the observations of Smith (2010).

## 5   Conclusions

In the current study, we investigated for the first time the potential of thrust set-point optimization in large wind farms for mitigating gravity-wave induced blockage effects, with the aim of increasing the wind-farm energy extraction. Thus, a fast boundary layer model proposed by Allaerts and Meyers (2019) was adopted. The three-layer model simulates the atmospheric response to turbine drag in large wind farms by dividing the vertical structure of the atmosphere into three layers. This approach accurately captures the effects of regional pressure gradients induced by large wind farms at low computational expenses. We first added the time-dependency to the model so that time-periodic gravity-wave patterns could be reproduced. Further, we reformulated the model as an optimization framework with the objective of maximizing the wind-farm energy output at all costs. Gradient information was derived using the continuous adjoint method. To limit the computational cost, a box-function wind-farm force model was used which assumes that the force is distributed over the whole wind-farm area. The wind-farm layout was inspired by the works of Allaerts et al. (2018) and Allaerts and Meyers (2019), roughly representing the Belgian–Dutch offshore wind-farm cluster.

The optimization model was applied to two different atmospheric states representative of subcritical ($F_r = 0.9$) and supercritical ($F_r = 1.1$) flow conditions. The optimal configurations were then compared with a reference model setup which uses a uniform thrust-coefficient distribution. We did not observe dynamic behaviour in the optimal thrust set-point distributions for different choices of time horizon and time step, meaning that it is not necessary to excite non-stationary wave patterns to further increase the wind-farm energy output. However, we observed interesting spatial patterns. The optimal thrust set-point distributions turned out to be inversely related with the inversion-layer vertical displacement $\eta_t$. This has led to a sinusoidal and U-shaped $C_T^O$ distribution along the streamwise direction in sub- and supercritical conditions, respectively. An inversion-layer displacement reduction of 14.5% and 16.8% was observed in sub- and supercritical conditions, which lowered the adverse pressure gradient strength in the wind-farm induction and entrance region. The reduced blockage effects allowed for higher flow wind speeds through the farm. The optimal configurations showed power gains of 5.3% and 7% in sub- and supercritical conditions with respect to the reference model setup.

The atmospheric state is far from being constant in real case scenarios, therefore the power gain sensitivity to changes in atmospheric conditions was further studied. Thus, the developed thrust set-point optimization tool was applied for several wind profiles, inversion strengths and atmosphere stratifications for a total of 1960 different atmospheric states. Regions of high inversion-layer-displacement reduction in the sensitivity domain strictly corresponded to regions of high power gain. This has confirmed that it is essential to reduce the streamline divergence over the wind-farm area for limiting gravity-wave induced power loss. The strong gravity-wave feedback in high $P_N$ conditions made these atmospheric states the most suitable for power recovery purposes. Power gains up to 14% were found for weakly stratified atmospheres ($P_N \approx 11$) in correspondence of critical flow conditions ($F_r = 1$). This is related to the large flow perturbations induced by the chocking effect (Smith, 2010). Overall, power gains above 4% were observed for 77% of the cases. We note that the gravity-wave induced power losses are also sensitive to the wind-farm layout. Optimization of layout (including, e.g., relevant techno-economical constraints) is however not considered here and can be an interesting topic for future research.

The results discussed in the current manuscript make wind-farm set-point optimization a promising tool for gravity-wave induced power loss recovery. However, many challenges remain before this can be translated to real wind-farm applications. In the current work, we did not include an explicit wake model in our model, and have presumed that wake losses remain unchanged during optimization (i.e., $\eta_{\mathrm{w}}$ is assumed to be constant). In the future, an analytical wind-farm wake model, such as, e.g., the one developed by Niayifar and Porté-Agel (2016) and used by Allaerts and Meyers (2019) could be adopted for optimization. This would however also require better representation in the wake model of changing background variables and pressure gradients. For instance, gravity-wave induced pressure gradient effects on turbine wake recovery could be included using the model proposed by Shamsoddin and Porté-Agel (2018) that incorporates effects of pressure gradients. Furthermore, the use of a wind-farm drag model which computes analytically the wake of each turbine would allow us to investigate separately the influence of wake effects and gravity waves on the optimal turbine set-points. This is work for future research. Moreover, the three-layer model has been validate with LES results only (Allaerts and Meyers (2019)). We are planning to perform a more extensive validation of the model in the near future. Next, we also plan to apply the results obtained in this article to a higher fidelity model (i.e, our in-house LES solver SP-Wind). However, this requires some work on the efficiency of non-reflecting boundary conditions in our LES solver (Allaerts and Meyers, 2017, 2018). Finally, we assumed that the free atmosphere is uniformly stratified, and steady. The relaxation of these assumptions would extend the applicability of the model, e.g, to atmospheres with height-dependent Brunt-Väisälä frequency, and geostrophic wind, among others.

**Appendix A:  Derivation and verification of the adjoint equations and the adjoint gradient**

The continuous adjoint method is briefly explained in Appendix A1. Next, the three-layer model adjoint equations and cost functional gradient are derived in Appendix A2 and A3, respectively. Finally, the comparison between a finite difference approximation of the cost function gradient and the adjoint evaluation is performed in Appendix A4.

## A1 Continuous adjoint method

We adopt the standard $L^2$ inner product over the time interval $[0,T]$ and simulation domain $\Omega$

$$(\boldsymbol{a},\boldsymbol{b}) = \int_0^T \iint_\Omega \boldsymbol{a} \cdot \boldsymbol{b} \, \mathrm{d}\boldsymbol{x}\mathrm{d}t \tag{A1}$$

where $\boldsymbol{a}$ and $\boldsymbol{b}$ are two generic vectors. Moreover, we denote with $\boldsymbol{\psi} = [u_1, v_1, u_2, v_2, p_1, p_2]$ the vector containing the state variables and with $C_T = C_T(x,y,t)$ the control parameter.

The reduced cost functional is defined as

$$\widetilde{\mathcal{J}}(C_T) = \int_0^T \iint_\Omega \mathcal{K}\big(\boldsymbol{\psi}(C_T), C_T\big) \, \mathrm{d}\boldsymbol{x}\mathrm{d}t \tag{A2}$$

where

$$\mathcal{K}\big(\boldsymbol{\psi}(C_T), C_T\big) = -\beta \|\boldsymbol{U}_1\| C_p B(x,y) \Big( \|\boldsymbol{U}_1\|^2 + 3\boldsymbol{U}_1 \cdot \boldsymbol{u}_1 \Big). \tag{A3}$$

The gradient of the reduced cost functional $\nabla\widetilde{\mathcal{J}}$ is interpreted as the Riesz representation of the Gâteaux derivative operator at $C_T$ in any arbitrary direction $\delta C_T$

$$\widetilde{\mathcal{J}}_{C_T}(\delta C_T) \equiv \frac{\mathrm{d}}{\mathrm{d}\alpha} \widetilde{\mathcal{J}}(C_T + \alpha\delta C_T)\Big|_{\alpha=0} = \big(\nabla\widetilde{\mathcal{J}}, \delta C_T\big) \qquad \forall\, \delta C_T \in \mathcal{H} \tag{A4}$$

where $\mathcal{H}$ denotes the control Hilbert space.

Next, we define the state constraints of the optimization problem (i.e., the three-layer model equations) with shorthand notation $\mathcal{N}(\boldsymbol{\psi}, C_T)$. The reduced formulation of the optimization problem implies by definition that $\mathcal{N}\big(\boldsymbol{\psi}(C_T), C_T\big) = 0$, therefore we can write the reduced cost functional as

$$\widetilde{\mathcal{J}}(C_T) = \mathcal{J}\big(\boldsymbol{\psi}(C_T), C_T\big) + \Big(\boldsymbol{\psi}^*, \mathcal{N}\big(\boldsymbol{\psi}(C_T), C_T\big)\Big) \tag{A5}$$

where $\boldsymbol{\psi}^* = [\zeta_1, \chi_1, \zeta_2, \chi_2, \Pi_1, \Pi_2]$ denotes the vector containing the adjoint variables which play the role of Lagrange multipliers. In fact, it is easy to notice that $\widetilde{\mathcal{J}}(C_T) = \mathcal{L}\big(C_T, \boldsymbol{\psi}(C_T), \boldsymbol{\psi}^*\big)$, where $\mathcal{L}$ is the Lagrangian of the optimization problem in Eq. 16.

Using A4 and A5, the gradient of the reduced cost functional can be expressed as

$$\big(\nabla\widetilde{\mathcal{J}}, \delta C_T\big) = \Big(\frac{\partial\mathcal{K}}{\partial C_T}, \delta C_T\Big) + \Big(\boldsymbol{\psi}^*, \frac{\partial\mathcal{N}}{\partial C_T}\delta C_T\Big) + \Big(\frac{\partial\mathcal{K}}{\partial\boldsymbol{\psi}}, \delta\boldsymbol{\psi}\Big) + \Big(\boldsymbol{\psi}^*, \frac{\partial\mathcal{N}}{\partial\boldsymbol{\psi}}\delta\boldsymbol{\psi}\Big) \tag{A6}$$

where $\delta\boldsymbol{\psi} = \mathrm{d}\boldsymbol{\psi}/\mathrm{d}C_T \, \delta C_T$. The adjoint of the operator $\partial\mathcal{N}/\partial\boldsymbol{\psi}$ is given by

$$\Big(\boldsymbol{\psi}^*, \frac{\partial\mathcal{N}}{\partial\boldsymbol{\psi}}\delta\boldsymbol{\psi}\Big) = \Big(\Big[\frac{\partial\mathcal{N}}{\partial\boldsymbol{\psi}}\Big]^* \boldsymbol{\psi}^*, \delta\boldsymbol{\psi}\Big) + BT_1 \tag{A7}$$

where the right-hand side is found using integration by parts. Similarly, the adjoint of $\partial \mathcal{N}/\partial C_T$ is expressed as

$$\left(\boldsymbol{\psi}^*, \frac{\partial \mathcal{N}}{\partial C_T}\delta C_T\right) = \left(\left[\frac{\partial \mathcal{N}}{\partial C_T}\right]^*\boldsymbol{\psi}^*, \delta C_T\right) + BT_2. \tag{A8}$$

The boundary terms $BT_1$ and $BT_2$ arise as a result of the integration by parts. Due to spatial- and time-periodicity constraints, it is easy to show that $BT_1 = BT_2 = 0$. Hence, substituting A7 and A8 into A6, we obtain

$$(\nabla \tilde{\mathcal{J}}, \delta C_T) = \left(\frac{\partial \mathcal{K}}{\partial C_T} + \left[\frac{\partial \mathcal{N}}{\partial C_T}\right]^*\boldsymbol{\psi}^*, \delta C_T\right) + \left(\frac{\partial \mathcal{K}}{\partial \boldsymbol{\psi}} + \left[\frac{\partial \mathcal{N}}{\partial \boldsymbol{\psi}}\right]^*\boldsymbol{\psi}^*, \delta \boldsymbol{\psi}\right). \tag{A9}$$

Further, we assume that the adjoint variables satisfy the following relation

$$\left(\frac{\partial \mathcal{K}}{\partial \boldsymbol{\psi}} + \left[\frac{\partial \mathcal{N}}{\partial \boldsymbol{\psi}}\right]^*\boldsymbol{\psi}^*, \delta \boldsymbol{\psi}\right) = 0 \tag{A10}$$

which defines the adjoint equations. Therefore, the adjoint gradient is given by

$$\nabla \tilde{\mathcal{J}} = \frac{\partial \mathcal{K}}{\partial C_T} + \left[\frac{\partial \mathcal{N}}{\partial C_T}\right]^*\boldsymbol{\psi}^*. \tag{A11}$$

## A2    Derivation of the adjoint equations

We apply relation A7 for deriving the adjoint of the operator $\partial \mathcal{N}/\partial \boldsymbol{\psi}$. Starting with the velocity perturbations in the wind-farm layer, we have

$$\left(\boldsymbol{\psi}^*, \frac{\partial \mathcal{N}}{\partial \boldsymbol{u}_1}\delta \boldsymbol{u}_1\right) = \int\limits_0^T \iint\limits_\Omega \left[\frac{\partial \delta \boldsymbol{u}_1}{\partial t} + \boldsymbol{U}_1 \cdot \nabla \delta \boldsymbol{u}_1 + f_c \boldsymbol{J} \cdot \delta \boldsymbol{u}_1 - \nu_{t,1}\nabla^2 \delta \boldsymbol{u}_1 + \frac{\boldsymbol{D}'}{H_1} \cdot \delta \boldsymbol{u}_1 + \frac{\boldsymbol{C}'}{H_1} \cdot \delta \boldsymbol{u}_1 + \right.$$

$$\left. -\frac{1}{H_1}\frac{\partial \boldsymbol{f}^{(1)}}{\partial \boldsymbol{u}_1}\bigg|_{\delta \boldsymbol{u}_1}\right] \cdot \boldsymbol{\zeta}_1 \, \mathrm{d}\boldsymbol{x}\mathrm{d}t + \int\limits_0^T \iint\limits_\Omega \left[-\frac{\boldsymbol{D}'}{H_2} \cdot \delta \boldsymbol{u}_1\right] \cdot \boldsymbol{\zeta}_2 \, \mathrm{d}\boldsymbol{x}\mathrm{d}t + \int\limits_0^T \iint\limits_\Omega \left[H_1 \mathcal{F}^{-1}(\hat{\Phi}) * \nabla \cdot \delta \boldsymbol{u}_1\right]\Pi_1 \, \mathrm{d}\boldsymbol{x}\mathrm{d}t \tag{A12}$$

and by computing an integration by parts we obtain

$$\left(\left[\frac{\partial \mathcal{N}}{\partial \boldsymbol{u}_1}\right]^*\boldsymbol{\psi}^*, \delta \boldsymbol{u}_1\right) = \int\limits_0^T \iint\limits_\Omega \left[-\frac{\partial \boldsymbol{\zeta}_1}{\partial t} - \boldsymbol{U}_1 \cdot \nabla \boldsymbol{\zeta}_1 + f_c \boldsymbol{J} \cdot \boldsymbol{\zeta}_1 - \nu_{t,1}\nabla^2 \boldsymbol{\zeta}_1 + \frac{\boldsymbol{D}'}{H_1} \cdot \boldsymbol{\zeta}_1 + \frac{\boldsymbol{C}'}{H_1} \cdot \boldsymbol{\zeta}_1 + \right.$$

$$\left. + \frac{\beta C_T B(x,y)}{H_1}\boldsymbol{U}' \cdot \boldsymbol{\zeta}_1 - \frac{\boldsymbol{D}'}{H_2} \cdot \boldsymbol{\zeta}_2 - H_1\left[\mathcal{F}^{-1}(\hat{\Phi})(-\boldsymbol{x}, -t) * \nabla \Pi_1\right]\right] \cdot \delta \boldsymbol{u}_1 \, \mathrm{d}\boldsymbol{x}\mathrm{d}t. \tag{A13}$$

Note that the minus sign in the argument of $\mathcal{F}^{-1}(\hat{\Phi})(-\boldsymbol{x}, -t)$ does not come from classical integration by parts. In fact, given three functions $f, g, h \in L^1(\Omega)$, it can be shown that

$$\int\limits_\Omega \left[f(x) * g(x)\right]h(x)dx = \int\limits_\Omega \int\limits_{\Omega'} \left[f(x - x')g(x')dx'\right]h(x)dx$$

$$= \int\limits_{\Omega'} \int\limits_\Omega f(-(x' - x))h(x)dxg(x')dx'$$

$$= \int\limits_\Omega \left[f(-x) * h(x)\right]g(x)dx. \tag{A14}$$

where in the second passage we have changed the order of integration (Fubini's theorem). This property allows us to write

$$-H_1 \int_0^T \iint_\Omega \left[ \mathcal{F}^{-1}(\hat{\Phi}) * \delta \boldsymbol{u}_1 \right] \cdot \nabla \Pi_1 \, \mathrm{d}\boldsymbol{x}\mathrm{d}t = -H_1 \int_0^T \iint_\Omega \left[ \mathcal{F}^{-1}(\hat{\Phi})(-\boldsymbol{x}, -t) * \nabla \Pi_1 \right] \cdot \delta \boldsymbol{u}_1 \, \mathrm{d}\boldsymbol{x}\mathrm{d}t. \tag{A15}$$

Similarly, for the velocity perturbations in the upper layer, we have that

$$\left( \left[ \frac{\partial \mathcal{N}}{\partial \boldsymbol{u}_2} \right]^* \boldsymbol{\psi}^*, \delta \boldsymbol{u}_2 \right) = \int_0^T \iint_\Omega \left[ -\frac{\boldsymbol{D}'}{H_1} \cdot \boldsymbol{\zeta_1} - \frac{\partial \boldsymbol{\zeta}_2}{\partial t} - \boldsymbol{U}_2 \cdot \nabla \boldsymbol{\zeta}_2 + f_c \boldsymbol{J} \cdot \boldsymbol{\zeta}_2 - \nu_{t,2} \nabla^2 \boldsymbol{\zeta}_2 + \frac{\boldsymbol{D}'}{H_2} \cdot \boldsymbol{\zeta}_2 + \right.$$
$$\left. - H_2 \left[ \mathcal{F}^{-1}(\hat{\Phi})(-\boldsymbol{x}, -t) * \nabla \Pi_2 \right] \right] \cdot \delta \boldsymbol{u}_2 \, \mathrm{d}\boldsymbol{x}\mathrm{d}t. \tag{A16}$$

Following the same procedure for the pressure perturbations $p_1$ and $p_2$, we obtain

$$\left( \left[ \frac{\partial \mathcal{N}}{\partial p_1} \right]^* \boldsymbol{\psi}^*, \delta p_1 \right) = \int_0^T \iint_\Omega \left[ -\frac{1}{\rho_0} \nabla \cdot \boldsymbol{\zeta}_1 - \frac{1}{\rho_0} \nabla \cdot \boldsymbol{\zeta}_2 - \frac{1}{\rho_0} \frac{\partial \Pi_1}{\partial t} - \frac{1}{\rho_0} \boldsymbol{U}_1 \cdot \nabla \Pi_1 \right] \delta p_1 \, \mathrm{d}\boldsymbol{x}\mathrm{d}t \tag{A17}$$

and

$$\left( \left[ \frac{\partial \mathcal{N}}{\partial p_2} \right]^* \boldsymbol{\psi}^*, \delta p_2 \right) = \int_0^T \iint_\Omega \left[ -\frac{1}{\rho_0} \nabla \cdot \boldsymbol{\zeta}_1 - \frac{1}{\rho_0} \nabla \cdot \boldsymbol{\zeta}_2 - \frac{1}{\rho_0} \frac{\partial \Pi_2}{\partial t} - \frac{1}{\rho_0} \boldsymbol{U}_2 \cdot \nabla \Pi_2 \right] \delta p_2 \, \mathrm{d}\boldsymbol{x}\mathrm{d}t. \tag{A18}$$

Using A10, the resulting adjoint equations correspond to

$$-\frac{\partial \boldsymbol{\zeta}_1}{\partial t} - \boldsymbol{U}_1 \cdot \nabla \boldsymbol{\zeta}_1 + f_c \boldsymbol{J} \cdot \boldsymbol{\zeta}_1 - \nu_{t,1} \nabla^2 \boldsymbol{\zeta}_1 + \frac{\boldsymbol{D}'}{H_1} \cdot \boldsymbol{\zeta}_1 + \frac{\boldsymbol{C}'}{H_1} \cdot \boldsymbol{\zeta}_1 - \frac{\boldsymbol{D}'}{H_2} \cdot \boldsymbol{\zeta}_2 - H_1 \left[ \mathcal{F}^{-1}(\hat{\Phi})(-\boldsymbol{x}, -t) * \nabla \Pi_1 \right] +$$
$$+ \frac{\beta C_T B(x,y)}{H_1} \boldsymbol{U}' \cdot \boldsymbol{\zeta}_1 = -\frac{\partial \mathcal{K}}{\partial \boldsymbol{u}_1} \qquad\qquad \text{in } \Omega \times (0, T],$$

$$-\frac{\partial \boldsymbol{\zeta}_2}{\partial t} - \boldsymbol{U}_2 \cdot \nabla \boldsymbol{\zeta}_2 + f_c \boldsymbol{J} \cdot \boldsymbol{\zeta}_2 - \nu_{t,2} \nabla^2 \boldsymbol{\zeta}_2 + \frac{\boldsymbol{D}'}{H_2} \cdot \boldsymbol{\zeta}_2 - \frac{\boldsymbol{D}'}{H_1} \cdot \boldsymbol{\zeta}_1 - H_2 \left[ \mathcal{F}^{-1}(\hat{\Phi})(-\boldsymbol{x}, -t) * \nabla \Pi_2 \right] = -\frac{\partial \mathcal{K}}{\partial \boldsymbol{u}_2} \quad \text{in } \Omega \times (0, T],$$

$$-\frac{\partial \Pi_1}{\partial t} - \boldsymbol{U}_1 \cdot \nabla \Pi_1 - \nabla \cdot \boldsymbol{\zeta}_1 - \nabla \cdot \boldsymbol{\zeta}_2 = -\rho_0 \frac{\partial \mathcal{K}}{\partial p_1} \qquad\qquad \text{in } \Omega \times (0, T],$$

$$-\frac{\partial \Pi_2}{\partial t} - \boldsymbol{U}_2 \cdot \nabla \Pi_2 - \nabla \cdot \boldsymbol{\zeta}_1 - \nabla \cdot \boldsymbol{\zeta}_2 = -\rho_0 \frac{\partial \mathcal{K}}{\partial p_2} \qquad\qquad \text{in } \Omega \times (0, T].$$

$$\tag{A19}$$

The adjoint momentum equations of the upper layer are homogeneous, since the adjoint wind-farm drag force is felt only indirectly in this layer ($\partial \mathcal{K} / \partial \boldsymbol{u}_2 = 0$). Moreover, also $\partial \mathcal{K} / \partial p_1 = \partial \mathcal{K} / \partial p_2 = 0$. On the other hand, the adjoint momentum equations of the wind-farm layer are driven by the cost function. Using A3, we obtain

$$\frac{\partial \mathcal{K}}{\partial \boldsymbol{u}_1} = -3\beta C_p B(x,y) \|\boldsymbol{U}_1\| \boldsymbol{U}_1. \tag{A20}$$

Fig. A1 illustrates a planform view of the forward and adjoint solutions in subcritical flow conditions ($F_r = 0.9$). Both solutions are derived assuming a steady-state formulation of the optimization problem. The numerical setup, wind-farm layout and

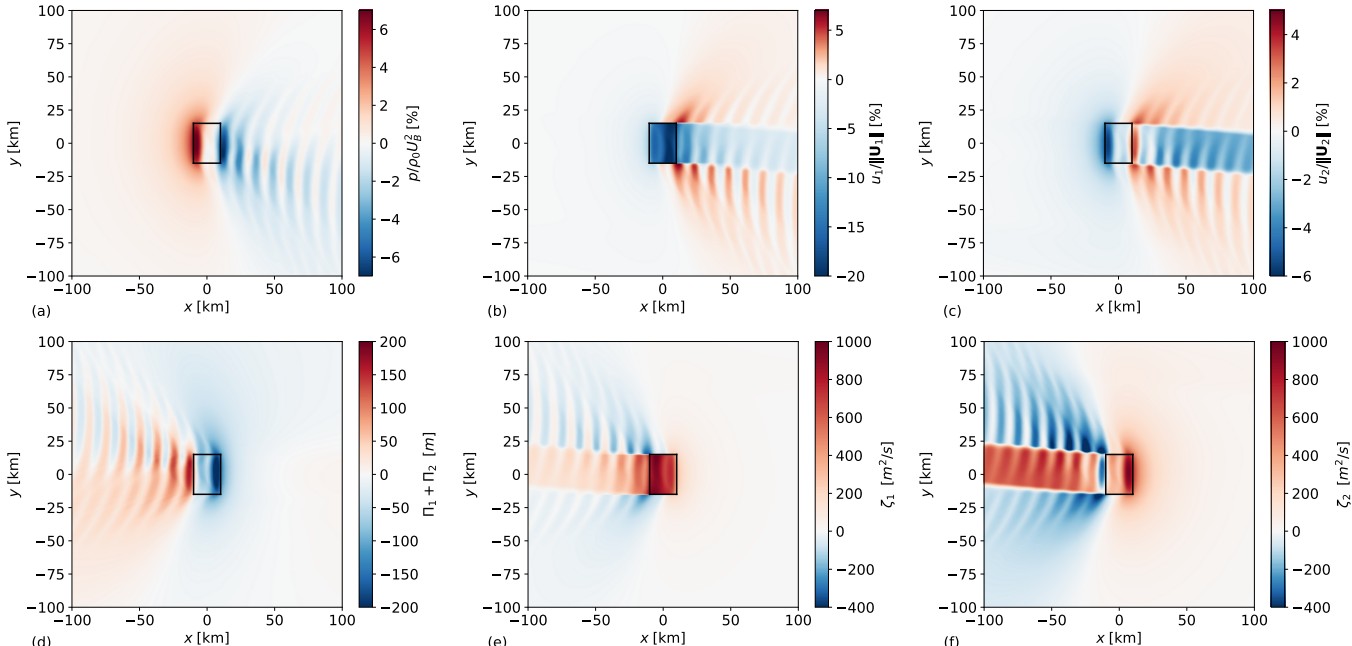

**Figure A1.** Planform view of (a) pressure perturbation, (b) velocity perturbation in the wind-farm layer, (c) velocity perturbation in the upper layer, (d) adjoint pressure $\Pi = \Pi_1 + \Pi_2$, (e) adjoint velocity field in the wind-farm layer and (f) adjoint velocity field in the upper layer in subcritical ($F_r = 0.9$) flow conditions. The black rectangle indicates the wind-farm region.

atmospheric state are the ones listed in Table 1. Due to integration by parts, the convective term is negative in the backward equations, causing the flow to propagate upstream (i.e., from right to left of our domain) as displayed in Fig. A1 (bottom row). Moreover, the wind farm acts as a source term and it speeds up the adjoint solution instead of decelerating it, causing an acceleration within the wind-farm area and in the wake region.

## A3 Derivation of the gradient

The adjoint gradient of the cost function is derived using relation A11. To compute the adjoint of the operator $\partial \mathcal{N} / \partial C_T$, we need to evaluate the following inner product

$$
\begin{aligned}
\left(\boldsymbol{\psi}^*, \frac{\partial \mathcal{N}}{\partial C_T} \delta C_T\right) &= \int_0^T \iint_\Omega \left[ -\frac{1}{H_1} \frac{\partial \boldsymbol{f}^{(0)}}{\partial C_T} \bigg|_{\delta C_T} - \frac{1}{H_1} \frac{\partial \boldsymbol{f}^{(1)}}{\partial C_T} \bigg|_{\delta C_T} \right] \cdot \boldsymbol{\zeta_1} \, \mathrm{d}\boldsymbol{x}\mathrm{d}t \\
&= \int_0^T \iint_\Omega \left[ \frac{\beta B(x,y)}{H_1} \delta C_T \|\boldsymbol{U}_1\|\boldsymbol{U}_1 + \frac{\beta B(x,y)}{H_1} \delta C_T \boldsymbol{U}' \cdot \boldsymbol{u}_1 \right] \cdot \boldsymbol{\zeta}_1 \, \mathrm{d}\boldsymbol{x}\mathrm{d}t
\end{aligned}
\tag{A21}
$$

which is easily rewritten as

$$\left(\left[\frac{\partial \mathcal{N}}{\partial C_T}\right]^* \boldsymbol{\psi}^*, \delta C_T\right) = \int\limits_0^T \iint\limits_\Omega \left[\frac{\beta B(x,y)}{H_1}\left(\|\boldsymbol{U}_1\|\boldsymbol{U}_1 \cdot \boldsymbol{\zeta_1} + \boldsymbol{u}_1^\top \cdot \boldsymbol{U}' \cdot \boldsymbol{\zeta_1}\right)\right]\delta C_T \; \mathrm{d}\boldsymbol{x}\mathrm{d}t. \tag{A22}$$

Moreover, we derive the first term on the right-hand side of A11 using A3, which results in

$$\frac{\partial \mathcal{K}}{\partial C_T} = -\beta B(x,y)\|\boldsymbol{U}_1\|\frac{\mathrm{d}C_p}{\mathrm{d}C_T}\left(\|\boldsymbol{U}_1\|^2 + 3\boldsymbol{U}_1 \cdot \boldsymbol{u}_1\right). \tag{A23}$$

Finally, we obtain the gradient expression by substituting A22 and A23 in A11, which gives

$$\nabla\widetilde{\mathcal{J}} = \frac{\beta B(x,y)}{H_1}\left[\|\boldsymbol{U}_1\|\boldsymbol{U}_1 \cdot \boldsymbol{\zeta_1} - H_1\|\boldsymbol{U}_1\|\frac{\mathrm{d}C_p}{\mathrm{d}C_T}\left(\|\boldsymbol{U}_1\|^2 + 3\boldsymbol{U}_1 \cdot \boldsymbol{u}_1\right) + \boldsymbol{u}_1^\top \cdot \boldsymbol{U}' \cdot \boldsymbol{\zeta_1}\right]. \tag{A24}$$

### A4   Verification of the adjoint gradient

The aim of this paragraph is to asses the quality of the gradient through comparison with a finite difference approximation. The comparison is done using a grid resolution of 500 m. All other parameters correspond to the ones listed in Table 1, with $F_r = 0.9$.

We define with

$$\nabla\widetilde{\mathcal{J}}_{\mathrm{ADJ}} = \left(\nabla\widetilde{\mathcal{J}}, \delta C_T\right) \tag{A25}$$

the directional derivative of $\nabla\widetilde{\mathcal{J}}$ along $\delta C_T$, where $\nabla\widetilde{\mathcal{J}}$ is the gradient computed with A24 and $\delta C_T$ is a perturbation of the baseline control $C_T$. Using finite difference, the same directional derivative can be approximated as

$$\nabla\widetilde{\mathcal{J}}_{\mathrm{FD}} = \frac{\widetilde{\mathcal{J}}\left(C_T + \alpha\delta C_T\right) - \widetilde{\mathcal{J}}\left(C_T\right)}{\alpha} + \mathcal{O}(\alpha). \tag{A26}$$

The truncation error of A26 is proportional to the order of magnitude of the step length $\alpha$. Therefore, $\alpha$ should be as small as possible to limit the discretization error. However, small values of $\alpha$ induce round-off errors due to finite-precision floating-point arithmetic. In other words, relation A26 provides accurate gradient information only for a lower an upper bounded range of step length values.

Next, we define

$$R = \frac{\nabla\widetilde{\mathcal{J}}_{\mathrm{ADJ}}}{\nabla\widetilde{\mathcal{J}}_{\mathrm{FD}}}, \tag{A27}$$

$$\mathcal{E} = \left|\frac{\nabla\widetilde{\mathcal{J}}_{\mathrm{ADJ}} - \nabla\widetilde{\mathcal{J}}_{\mathrm{FD}}}{\nabla\widetilde{\mathcal{J}}_{\mathrm{FD}}}\right| \tag{A28}$$

where $R$ and $\mathcal{E}$ represent the ratio and the relative error between gradient information computed with the adjoint and finite difference method. If the continuous adjoint method provides correct gradient information, we expect $R \simeq 1$ and $\mathcal{E}$ to be sufficiently small.

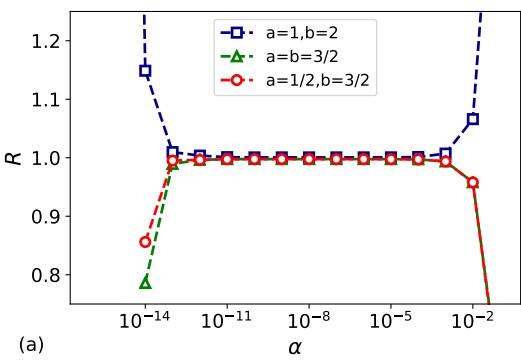 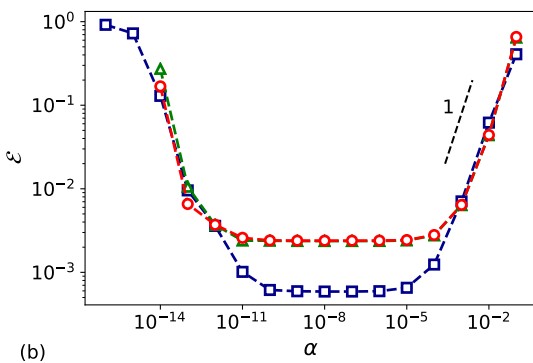

(a)  (b)

**Figure A2.** (a) Ratio and (b) relative error between adjoint and finite-difference based gradient.

The following generic baseline control is chosen

$$C_T^{\mathrm{B}}(x,y) = C_T^{\mathrm{Betz}}\left[\frac{1}{2} + \frac{1}{5}\cos\left(k_x x + \pi\right) + \frac{1}{5}\sin\left(k_y y + \pi/5\right)\right] \tag{A29}$$

where $C_T^{\mathrm{Betz}} = 8/9$, $k_x = 2\pi/L_x$ and $k_y = 2\pi/L_y$. Ideally, we should validate the adjoint-based gradient against the finite-
difference one for all possible perturbations $\delta C_T$. However, such validation would require to solve the governing equations
(forward and backward) $2.4 \times 10^3$ times since the control space has such DOF using this numerical setup. This computation is
too expensive, therefore we select a limited class of perturbations given by

$$\delta C_T(x,y) = \cos\left(a k_x x + \pi\right) + \sin\left(b k_y y + \pi/5\right) \tag{A30}$$

for different values of $a$ and $b$.

Results of the comparison are shown in Fig. A2. We can appreciate that for $10^{-11} \leq \alpha \leq 10^{-4}$ the ratio $R$ is very close
to unity and the relative error $\mathcal{E}$ is in the order of $10^{-4}$, showing the typical U-shaped curve (Nita et al., 2016). However, for
smaller step length values the relative error increases due to the decreasing arithmetic accuracy of the finite-difference based
gradient. The relative error also increases for $\alpha > 10^{-4}$ due to discretization errors. We can appreciate that Fig. A2b displays
a first-order truncation error in accordance with relation A26.

**Appendix B:  Optimal uniform thrust set-point distribution**

In the current section, we use the optimization framework derived in Section 2.2 to find an optimal uniform and steady thrust-
coefficient distribution that minimizes the gravity-wave induced blockage effects. To avoid confusion, we will denote with $C_T^{\mathrm{O}}$
and $C_T^{\mathrm{O,u}}$ the optimal non-uniform and uniform distribution, respectively. The wind-farm layout and the atmospheric state are
the ones detailed in Section 3.
Figure B1(a,b) displays the optimal spatially invariant $C_T^{\mathrm{O,u}}$ together with the streamwise profile of $C_T^{\mathrm{O}}$ through the center
of the farm, and its averaged value over the wind-farm area $\langle C_T^{\mathrm{O}}\rangle$ for the sub- and supercritical case, respectively. Moreover,

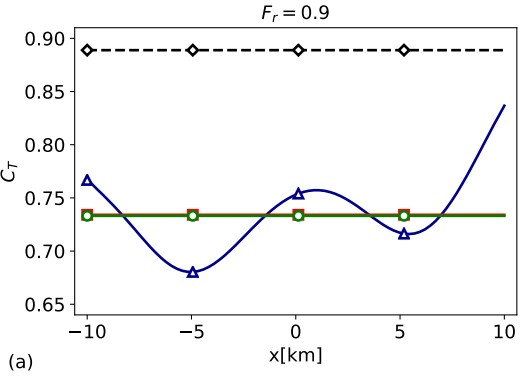 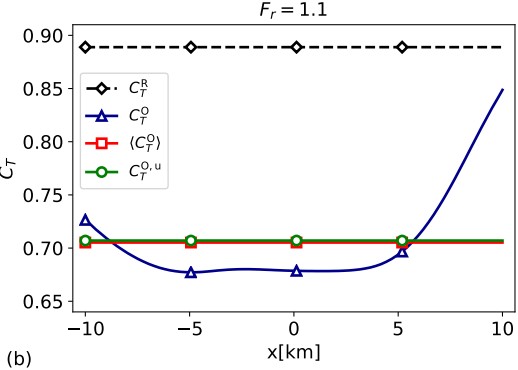

(a) (b)

**Figure B1.** Reference thrust set-point ($C_T^{\mathrm{R}}$), optimal non-uniform thrust set-point ($C_T^{\mathrm{O}}$) and its averaged value over the wind-farm area ($\langle C_T^{\mathrm{O}} \rangle$) and optimal uniform thrust coefficient distribution ($C_T^{\mathrm{O,u}}$) in (a) subcritical and (b) supercritical flow conditions. The $C_T^{\mathrm{O}}$ profiles are taken through the center of the farm ($y = 0$).

$C_T^{\mathrm{R}}$ denotes the thrust distribution used in the reference model. Interestingly, $C_T^{\mathrm{O,u}}$ corresponds to the average of the non-uniform distribution in both cases. Since $C_T^{\mathrm{O}}$ is sensitive to the atmospheric conditions, we expect $C_T^{\mathrm{O,u}}$ to depend as well on the atmospheric state (in fact, we observe a different value of $C_T^{\mathrm{O,u}}$ in sub- and supercritical conditions).

In the current example, the power gain $\mathcal{G}$ (see Eq. 21) over the reference model configuration obtained with the non-uniform distributions $C_T^{\mathrm{O}}$ are 5.3% and 7% for the sub- and supercritical case, respectively. For the optimal uniform distributions, we obtain a power gain of 5% and 6.6%.

## Appendix C: Grid sensitivity

A grid sensitivity analysis is performed to determine the dependence of the optimization results on the grid cell size. To this end, we fix the size of the numerical domain to $1000H \times 400H$ and we vary the grid resolution spanning from $5H$ to $H/3$, or equivalently from $1.6 \times 10^4$ to $3.6 \times 10^6$ DOF per layer. The results obtained are compared with the ones derived on a finer grid with resolution equal to $H/4$.

Fig. C1a and Fig. C1b display the cost function and power gain relative error, respectively, which are computed as

$$\mathcal{E}_{\mathcal{P}} = \left| \frac{\mathcal{P}^{\mathrm{F}} - \mathcal{P}}{\mathcal{P}} \right|, \tag{C1}$$

$$\mathcal{E}_{\mathcal{G}} = \left| \frac{\mathcal{G}^{\mathrm{F}} - \mathcal{G}}{\mathcal{G}} \right| \tag{C2}$$

where $\mathcal{P}^{\mathrm{F}} = \tilde{\mathcal{J}}^{\mathrm{F}}/T$ and $\mathcal{G}^{\mathrm{F}}$ are the cost function (scaled with the time horizon $T$) and power gain obtained with a $H/4$ grid resolution while $\mathcal{P} = \tilde{\mathcal{J}}/T$ and $\mathcal{G}$ are the ones obtained with coarser grids. Note that the optimal distributions are steady-state, therefore $\mathcal{E}_{\mathcal{P}}$ is not dependent on the choice of the time horizon $T$. The cost function is evaluated using the reference case setup. The power gain is obtained using the optimization model described in Section 2. The model setup is reported in Table 1.

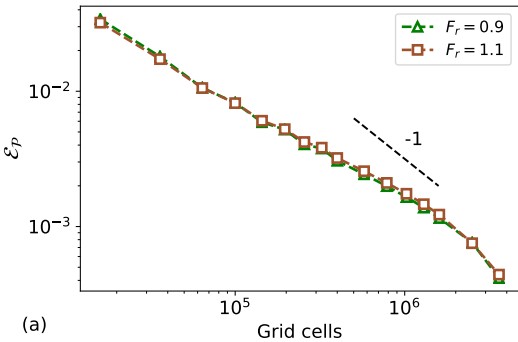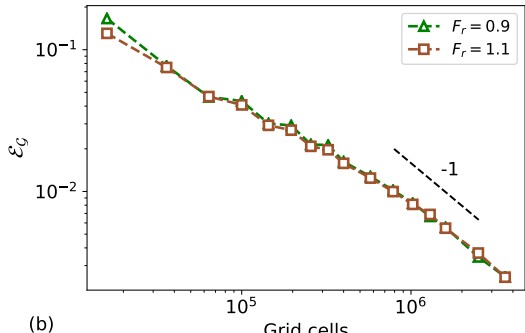

**Figure C1.** (a) Cost function and (b) power gain relative error between a grid with resolution $H/4$ and coarser grids in sub- and supercritical flow conditions.

Spectral methods are known to have exponential convergence when used for discretizing smooth functions (i.e., $f \in C^\infty$). However, algebraic convergence is attained for functions $f \in C^p$ with $p \geq 0$. Fig. C1 illustrates that we obtain a first-order convergence. This is due to the two-dimensional Heaviside function $B(x,y)$ used for representing the wind-farm footprint, which is discontinuous with discontinuous derivatives. Fig. C1 also confirms that the results of the optimization model are grid-independent. In fact, the cost function and power gain values change of about 1% and 4% when the number of grid cells is increased by two orders of magnitude (from $10^4$ to $10^6$). This justifies the use of a coarser grid in the sensitivity study performed in section 4.2.

*Author contributions.* L.L. and J.M. jointly set up the simulation studies in the current work. L.L. performed code implementations and carried out the simulations. L.L. and J.M. jointly wrote the manuscript.

*Competing interests.* The authors declare that they have no conflict of interest.

*Acknowledgements.* The authors acknowledge support from the Research Foundation Flanders (FWO, grant no. G0B1518N). The authors thank Prof. Nicole Van Lipzig for useful discussions. The computational resources and services in this work were provided by the VSC (Flemish Supercomputer Center), funded by the Research Foundation Flanders (FWO) and the Flemish Government department EWI.

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
