# Peer review of "Set-point optimization in wind farms to mitigate effects of flow blockage induced by atmospheric gravity waves"

_Wind Energy Science, 2020_

## Referee Comment (RC1) · Dries Allaerts (Referee) · 4 Jun 2020

This paper aims to identify the optimal distribution of wind turbine set-points to mitigate flow blockage induced by atmospheric gravity waves and hence maximize wind-farm energy extraction. The authors simulate the response of the atmospheric flow to the wind-farm drag using a recently developed mid-fidelity model, and they introduce a corresponding optimization framework based on the continuous adjoint method. The results are promising and show that a non-uniform spatial distribution of wind turbine set points can increase the energy extraction of the farm by reducing the excitation of atmospheric gravity waves. I believe this paper is of interest to the wind energy

community as it demonstrates the use of set-point optimization for wind farms and highlights the potential for new optimization and control strategies to cope with wind-farm scale blockage. I do have some scientific questions and technical comments as listed below.

Scientific comments/questions

1. Line 91: In the derivation of the three-layer model, no assumptions need to be made about the vertical component of the velocity. Rather, by averaging over the height of the respective layers, the horizontal momentum equations become independent of the vertical velocity.

2. Line 130: The relation between pressure and inversion layer displacement based on the complex stratification coefficient is not due to Gill 1982 (at least not the part concerning the atmospheric gravity waves). I think it is more appropriate to cite Smith 2010 instead.

3. Eq. 17: Can you elaborate on the function of the complex stratification coefficient in the adjoint equations? That is, what do you mean with the negation of the arguments x and t. I assume this arrives from the partial integration and is similar to the sign reversal of the convective term, but it is not clear to me how I should interpret the current notation.

4. Line 235: Can you comment on the numerical resources (time and number of processors if parallelized) it takes to compute an optimal set-point distribution? I am asking because a possible application could be using weather forecasts to update the set-point distribution when gravity waves are to be expected (e.g., forecast predicts shallow boundary layers in the next few hours).

5. Line 241: How did you select the relative turbine spacing?

6. Line 301: Favourable pressure gradients are also present in the bulk of the wind-farm, whereas the velocity deficits continue to increase throughout the farm and only

recover behind the farm. I believe the favourable pressure gradients do not necessarily accelerate the flow, but are instead balanced by a higher thrust force. Physically, this would correspond to the favourable pressure gradient re-energizing the wake flows and thereby reducing the turbine losses in the bulk of the farm. Can you comment on this?

7. Section 4.1: Did you consider optimizing for a uniform set-point distribution? What are the maximum gains to be expected there, and hence how much more is there to be gained by using a non-uniform set-point distribution? How would that uniform value compare to the average of the non-uniform distribution, and would the uniform value depend on the atmospheric condition as well?

8. Section 4.1: How does the power performance of the optimal set-point distribution compare to the idealized power output when all turbines would be operating in isolated conditions? I.e. how much of the power loss due to flow blockage is irreversible?

Technical comments

* Line 389: Typo in "dispersive"

* Label A12 and A13 reference the same equation split over two lines.

* Line 573: Typo in "through"

---

## Referee Comment (RC2) · Anonymous Referee #2 · 8 Jun 2020

Interesting paper and concept. Contains every detail of the simulation but requires a solid background in fluid dynamics to understand. The latter is directly a disclaimer since I don't have a solid background in this field. While reading the paper I have the following comments:

1) The authors state on pg. 11 line 280: "However, all optimal . . ... are constant in time. . . .. And conclude that unsteady time-periodic excitation is less effective" I believe that this claim is too strong. There is no guarantee that the optimizer will find the optimal solution. The optimizer finds a solution under the specified constraints, and that is what the authors present. This only has value if this gives rise to a better under-

standing of the physics. Now it is one of many possible solutions (local minima). The authors should consider rephrasing these claims to the assumptions made throughout the derivation.

2) The significance of the paper is also a bit unclear. In the conclusion the authors state that an optimization model was applied for set-point optimization. Many approximations have been made in the modelling step and there is no quantification of the potential error. The energy gains mentioned in the abstract are incredible high. I would like to see a validation of the model or the results applied to a high(er) validity model.

3) Generally, assumptions should be stated clearly. For example, the wakes between the turbines are not explicitly modelled. This is a large assumption to make, and is only briefly mentioned in the text. What is the expected impact on the results? How does it affect the conclusions drawn in the article? Also, the sampling time seems rather large for typical wind farm control algorithms. How does this impact your results? Would you be able to find a periodic optimal signal if you had a shorter sampling time? How about the fidelity of your rotor model – would things change with an ALM model?

4) The article is long, making it cumbersome to read. Perhaps certain parts can be omitted. For example, is the model from section 2.1 a novel contribution or is it identical to the one described in Allaerts & Meyers 2019? If the latter, consider removing it from this article.

5) Figure 1: It seems as if you have very few iterations before convergence. Can you comment on this?

---

## Referee Comment (RC3) · Alan Wai Hou Lio (Referee) · 15 Jun 2020

The paper is interesting. The authors investigated the problems of the mesoscale interaction between a wind farm and atmospheric boundary-layer. A three-layer model is proposed for modelling the wind-farm induced gravity wave. Based on the simplified model, optimisation is then proposed to find the optimal thrust coefficient distributions that maximise the wind farm power output. The concept is appealing and the topic is definitely relevant to the wind energy community. However, I found the paper is a bit too long (30 pages) and written for audiences with a strong background in fluid dynamics. As a disclaimer, my background is not in fluid dynamics. Please find my comments as

follows.

Comments: pg1: Some claims by the authors were not clear. For example, the optimal thrust coefficient distributions are spatially stationary rather than time-periodic. Did the authors consider turbulent wind inflow and turbine-to-turbine interactions? Does the claim imply that stationary spatial distributions of thrust coefficients are better than dynamically changing the thrust set-points for maximising the wind farm power? This claim disagreed with some of the other works (e.g. [1]). In [1], the benefit of periodic dynamic induction control was shown, where the thrust coefficient of the upstream turbine was periodically adjusted to improve the downstream wind flow. How is this work related to [1]?

pg2 l59: "asses" -> asks.

pg4 l116: Ct is a function of Ct(x,y,t)? What is x and y in B(x,y)?

pg5 l123: What is the dimension of Ct? Is Ct a vector where the number of elements in that vector is equal to the number of turbines? How is Ct of each turbine related the aggregate wind farm drag f?

p5 l124: " the thrust-coefficient distribution Ct has to be interpreted as a perturbation." Is Ct the thrust coefficient or the perturbation to the thrust coefficient?

p5 l151: " The goal of the optimization framework is to find a time-periodic optimal thrust-coefficient distribution". Why did the authors assume that the optimal thrust distribution would be time-periodic in the beginning?

p6: Equation (13), what is $\varphi$ and J in (13) is not a function of Ct. I suggest the authors swap equation (13) and (14) for clarity.

p7 l183: $\mathcal{N}(\varphi(C_t), C_t) = 0$. Is this only valid around the neighbourhood of the solution? What is $\mathcal{N}$?

p11 l268: what is $P_N$?

References:

[1] J. Frederik et al., "Periodic dynamic induction control of wind farms: proving the potential in simulations and wind tunnel experiments," Wind Energy Sci., no. August, pp. 1–18, 2019.

---

## Author Comment (AC1) · 27 Aug 2020

**Response to comments of Dries Allaerts**

- This paper aims to identify the optimal distribution of wind turbine set-points to mitigate flow blockage induced by atmospheric gravity waves and hence maximize wind-farm energy extraction. The authors simulate the response of the atmospheric flow to the wind-farm drag using a recently developed mid-fidelity model, and they introduce a corresponding optimization framework based on the continuous adjoint method. The results are promising and show that a non-uniform spatial distribution of wind turbine set points can increase the energy extraction of the farm by reducing the excitation of atmospheric gravity waves. I believe this paper is of interest to the wind energy community as it demonstrates the use of set-point optimization for wind farms and highlights the potential for new optimization and control strategies to cope with windfarm scale blockage.
  We would like to thank the referee for the kind words and the very constructive feedback in improving the quality of the paper.

- 1. Line 91: In the derivation of the three-layer model, no assumptions need to be made about the vertical component of the velocity. Rather, by averaging over the height of the respective layers, the horizontal momentum equations become independent of the vertical velocity.
  We have corrected our statement with the following sentences at P3-L90:

  "*The model equations are derived starting from the incompressible three-dimensional Reynolds-Averaged Navier-Stokes (RANS) equations for the ABL (Stull 1988). A depth-integration over the wind-farm and upper layer height is further computed, which removes the vertical velocity from the equations. Hence, the basic equation system is reduced to a set of only three equations: the continuity equation and the momentum equations in horizontal directions. Subsequently, the governing equations are linearized with respect to the background state variables, using some additional modelling assumptions for the turbulent stresses (see Allaerts and Meyers (2019) for more details).*"

- 2. Line 130: The relation between pressure and inversion layer displacement based on the complex stratification coefficient is not due to Gill 1982 (at least not the part concerning the atmospheric gravity waves). I think it is more appropriate to cite Smith 2010 instead.
  We agree on this. Hence, we have cited Smith 2010 instead of Gill 1982.

- 3. Eq. 17: Can you elaborate on the function of the complex stratification coefficient in the adjoint equations? That is, what do you mean with the negation of the arguments x and t. I assume this arrives from the partial integration and is similar to the sign reversal of the convective term, but it is not clear to me how I should interpret the current notation.
  The negation of the arguments x and t does not follow from partial integration. Rather, it follows from the following property. Given three functions $f, g, h \in L^1(\Omega)$, it can be shown

that

$$\int_\Omega \big[f(x)*g(x)\big]h(x)dx = \int_\Omega \int_{\Omega'} \big[f(x-x')g(x')dx'\big]h(x)dx$$

$$= \int_\Omega \int_{\Omega'} f(x-x')g(x')h(x)dx'dx$$

$$= \int_{\Omega'} \int_\Omega f(x-x')h(x)dxg(x')dx'$$

$$= \int_{\Omega'} \int_\Omega f(-(x'-x))h(x)dxg(x')dx'$$

$$= \int_\Omega \big[f(-x)*h(x)\big]g(x)dx.$$

Note that in the second passage we have changed the order of integration (Fubini's theorem). In the current application, this property allows us to write

$$-H_1 \int_0^T \iint_\Omega \Big[\mathcal{F}^{-1}(\hat{\Phi})*\delta\boldsymbol{u}_1\Big]\cdot\nabla\Pi_1 d\boldsymbol{x}dt = -H_1 \int_0^T \iint_\Omega \Big[\mathcal{F}^{-1}(\hat{\Phi})(-\boldsymbol{x},-t)*\nabla\Pi_1\Big]\cdot\delta\boldsymbol{u}_1 d\boldsymbol{x}dt.$$

To include this in the text, we have added the following sentence at P8-L213:

*"Note that the minus sign in the argument of $\mathcal{F}^{-1}(\hat{\Phi})(-\boldsymbol{x},-t)$ is not a result from classical integration by parts, but arrives from applying Fubini's theorem to the convolution term in Eq. 12 and Eq. 13 (see Appendix A3 for details)."*

Moreover, we have also added the following lines at P24-L581:

*"Note that the minus sign in the argument of $\mathcal{F}^{-1}(\hat{\Phi})(-\boldsymbol{x},-t)$ does not come from classical integration by parts. In fact, given three functions $f, g, h \in L^1(\Omega)$, it can be shown that*

$$\int_\Omega \big[f(x)*g(x)\big]h(x)dx = \int_\Omega \int_{\Omega'} \big[f(x-x')g(x')dx'\big]h(x)dx$$

$$= \int_{\Omega'} \int_\Omega f(-(x'-x))h(x)dxg(x')dx'$$

$$= \int_\Omega \big[f(-x)*h(x)\big]g(x)dx.$$

*where in the second passage we have changed the order of integration (Fubini's theorem). This property allows us to write*

$$-H_1 \int_0^T \iint_\Omega \Big[\mathcal{F}^{-1}(\hat{\Phi})*\delta\boldsymbol{u}_1\Big]\cdot\nabla\Pi_1 \, d\boldsymbol{x}dt = -H_1 \int_0^T \iint_\Omega \Big[\mathcal{F}^{-1}(\hat{\Phi})(-\boldsymbol{x},-t)*\nabla\Pi_1\Big]\cdot\delta\boldsymbol{u}_1 \, d\boldsymbol{x}dt.$$

*"*

- 4. Line 235: Can you comment on the numerical resources (time and number of processors if parallelized) it takes to compute an optimal set-point distribution? I am asking because a possible application could be using weather forecasts to update the set-point distribution when gravity waves are to be expected (e.g., forecast predicts shallow boundary layers in the next few hours).
  To include these information in the text, we have added the following paragraph at the end of section 3.1 (P10-L257):

  *"The solver (which is not parallelized) takes a couple of hours to solve the equations for*

*a grid with resolution of 250 m ($6.4 \times 10^6$ DOF per layer). Since convergence is reached after approximately 20 function evaluations (which means that we solve state and adjoint equations 20 times), the optimizer takes a couple of days to compute an optimal thrust set-point distribution. However, after this work was performed, we have upgraded the forward solver which is now 700 times faster than our previous version. Optimization of the backward solver is planned for the future, and we expect that this will lead to an optimization algorithm that will only take several minutes for the same case.*"

- 5. Line 241: How did you select the relative turbine spacing?

  In the study of Allaerts and Meyers (2019), the authors considered a rectangular shaped farm with length $L_x = 20$ km and width $L_y = 30$ km containing $N_t = 486$ turbines. These numbers where chosen to represent roughly the number of turbines and the area covered by the Belgian-Dutch wind-farm cluster. In our study, we select a wind-farm with the same dimensions. Moreover, to have a similar density of turbines in the farm, we fix the dimensionless turbine spacings to $s_x = s_y = 5.61$ so that $L_x L_y / s_x s_y D^2 \simeq 486$. To include this information, we have added the following sentence at P10-L267:

  "*The wind turbine relative spacings along the x- and y-direction are $s_x = s_y = 5.61$ (both non-dimensionalized with respect to the turbine rotor diameter $D$), so that the density of turbines in the farm is similar to the one of Allaerts and Meyers (2019) (i.e. leading to $\beta = 0.01$ in Eq. 8, setting both the wake efficiency $\eta_w$ and $\gamma$ to 0.9 as in Allaerts and Meyers (2018)). Note that we do not define a specific layout or a number of turbines but we only fix the density of turbines in the farm.*"

- 6. Line 301: Favourable pressure gradients are also present in the bulk of the wind farm, whereas the velocity deficits continue to increase throughout the farm and only recover behind the farm. I believe the favourable pressure gradients do not necessarily accelerate the flow, but are instead balanced by a higher thrust force. Physically, this would correspond to the favourable pressure gradient re-energizing the wake flows and thereby reducing the turbine losses in the bulk of the farm. Can you comment on this?

  We agree with the referee and we confirm that the sentence "*In both sub- and supercritical case, favourable pressure gradients develop within the wind-farm area which tend to accelerate the flow in the wind-farm exit region*" is misleading. In fact, favourable pressure gradients re-energize the waked flow reducing the velocity deficits in the bulk of the farm. This also partially explains the higher velocity deficits observed for the supercritical case over the wind-farm area if compared with the ones obtained in the subcritical case (the favourable pressure gradient is stronger in subcritical conditions than supercritical ones). Therefore, we have modified the sentence mentioned above to (P14-L341):

  "*In both sub- and supercritical case, favourable pressure gradients reduce the velocity deficits in the bulk of the farm.*"

- 7. Section 4.1: Did you consider optimizing for a uniform set-point distribution? What are the maximum gains to be expected there, and hence how much more is there to be gained by using a non-uniform set-point distribution? How would that uniform value compare to the average of the non-uniform distribution, and would the uniform value depend on the atmospheric condition as well?

  We thank the referee for the very insightful questions. To answer these questions, we have added the following section in the appendix (P28-L652):

  "*In the current section, we use the optimization framework derived in Section 2.2 to find an*

[Figure]

[Figure]

Figure 1: Reference thrust set-point $(C_T^{\mathrm{R}})$, optimal non-uniform thrust set-point $(C_T^{\mathrm{O}})$ and its averaged value over the wind-farm area $(\langle C_T^{\mathrm{O}} \rangle)$ and optimal uniform thrust coefficient distribution $(C_T^{\mathrm{O,u}})$ in (a) subcritical and (b) supercritical flow conditions. The $C_T^{\mathrm{O}}$ profiles are taken through the center of the farm $(y = 0)$.

*optimal uniform and steady thrust-coefficient distribution that minimizes the gravity-wave induced blockage effects. To avoid confusion, we will denote with $C_T^{\mathrm{O}}$ and $C_T^{\mathrm{O,u}}$ the optimal non-uniform and uniform distribution, respectively. The wind-farm layout and the atmospheric state are the ones detailed in Section 3.*

*Figure 1(a,b) displays the optimal spatially invariant $C_T^{\mathrm{O,u}}$ together with the streamwise profile of $C_T^{\mathrm{O}}$ through the center of the farm, and its averaged value over the wind-farm area $\langle C_T^{\mathrm{O}} \rangle$ for the sub- and supercritical case, respectively. Moreover, $C_T^{\mathrm{R}}$ denotes the thrust distribution used in the reference model. Interestingly, $C_T^{\mathrm{O,u}}$ corresponds to the average of the non-uniform distribution in both cases. Since $C_T^{\mathrm{O}}$ is sensitive to the atmospheric conditions, we expect $C_T^{\mathrm{O,u}}$ to depend as well on the atmospheric state (in fact, we observe a different value of $C_T^{\mathrm{O,u}}$ in sub- and supercritical conditions).*

*In the current example, the energy gain $\mathcal{G}$ (see Eq. 21 on article) over the reference model configuration obtained with the non-uniform distributions $C_T^{\mathrm{O}}$ are 5.3% and 7% for the sub- and supercritical case, respectively. For the optimal uniform distributions, we obtain an energy gain of 5% and 6.6%."*

- 8. Section 4.1: How does the power performance of the optimal set-point distribution compare to the idealized power output when all turbines would be operating in isolated conditions? I.e. how much of the power loss due to flow blockage is irreversible?

  The usage of a box-function model makes it difficult to answer this question. In fact, this model uniformly spreads the force over the simulation cells in the wind-farm area and does not represent the disturbances caused by each turbine in detail. Therefore, the concept of turbines operating in isolated conditions is not reproducible. What we can do is to consider "turbines" which operate in idealized conditions, by using a uniform thrust set-point distribution with $C_T = 0.88$. This is what we have defined as reference case and the energy gains are referred to this state. The comparison suggested by the referee would be realizable if analytical wake models would be used (such as, the Gaussian wake model proposed by Niayifar and Porté-Agel (2014)). In order to not overload the discussion in the manuscript, we decided to not further explicitly comment on this. However, the wind-farm force model used in the optimization solver will be improved in the future, as suggested in the conclusions, and this will allow us to answer this question in more detail.

- \* Line 389: Typo in "dispersive"
  \* Label A12 and A13 reference the same equation split over two lines.
  \* Line 573: Typo in "through"

Thank you, we have corrected these errata.

---

## Author Comment (AC2) · 27 Aug 2020

**Response to comments of Anonymous Referee #2**

- Interesting paper and concept. Contains every detail of the simulation but requires a solid background in fluid dynamics to understand.
  We would like to thank the referee for the constructive feedback in improving the quality of the paper.

- 1) The authors state on pg. 11 line 280: "However, all optimal.. ..are constant in time.. And conclude that unsteady time-periodic excitation is less effective" I believe that this claim is too strong. There is no guarantee that the optimizer will find the optimal solution. The optimizer finds a solution under the specified constraints, and that is what the authors present. This only has value if this gives rise to a better understanding of the physics. Now it is one of many possible solutions (local minima). The authors should consider rephrasing these claims to the assumptions made throughout the derivation.
  We fully agree with the referee's statement. Therefore, the following paragraph has been modified at P12-L307:

  "*The optimization model described in Section 2.2 is time and space dependent. Hence, the model is capable of finding a time-periodic optimal thrust-coefficient distribution over the wind-farm area in a fixed time interval $[0, T]$. However, all optimal thrust set-point distributions found for the different combinations of time horizons and time steps reported in Table 1, are constant in time. We have verified this using a range of steady and unsteady starting conditions for $C_T$ in the algorithm, but did not find any unsteady optimum. We believe that this is due to two reasons. Firstly, we use steady-state inflow conditions, therefore neglecting meso-scale temporal variations in the velocity field (these could lead to time-dependent optimal control signals, but are not included in the current work). Secondly, the objective function is non-convex and there is no proof about the uniqueness of global minima. Hence, there is no guarantee that the optimal solution found by the optimizer corresponds to a global optimum. Nevertheless, since we do not observe any unsteady behaviour in our optimal solutions, we show only steady-state results in the remainder of the manuscript, and conclude for the time being that unsteady time-periodic excitation is less effective than a stationary spatially optimal distribution in this context.*
  *We also note that our findings are in contrast with recent works of Goit and Meyers (2015), Munters and Meyers (2018) and Frederik et al. (2020), in which the authors illustrated the benefits of dynamic induction control over yaw and static induction control. However, the characteristic time scale of gravity-wave effects is estimated to be approximately 1 h (Gill 1982, Allaerts and Meyers 2019) which is an order of magnitude above the typical time scale of wake convection between turbines, and turbulent mixing in turbine wakes (this also justifies the larger sampling time used). Hence, while unsteadiness of the thrust coefficient (with a typical time scale of 50 seconds for large scale turbines) can lead to improved wake mixing (Goit and Meyers (2015), Munters and Meyers (2018), Frederik et al (2020)), it has no impact on phenomena that occur at larger time scales, such as wind-farm induced gravity waves.*"

- 2) The significance of the paper is also a bit unclear. In the conclusion the authors state that an optimization model was applied for set-point optimization. Many approximations have been made in the modelling step and there is no quantification of the potential error. The energy gains mentioned in the abstract are incredible high. I would like to see a validation of the model or the results applied to a high(er) validity model.
  We agree with the referee that we did not talk about the model validation. The constraints of our optimization model (the state equations) correspond to the same model derived by

Allaerts and Meyers (2019). This model has been validated in Allaerts and Meyers (2019) (see Section 3, VAL2) against LES results. The validation showed that the three-layer model outperforms the Smith (2010) model and agrees well with LES results for low perturbation values. To include this in the article, the following paragraph has been added at the end of section 2.1 (P6-L155):

"*The three-layer model configuration described above has been validated against LES results by Allaerts and Meyers (2019) (see Section 3 VAL2) on a two dimensional (x-z) domain (i.e., all spanwise derivatives are set to zero). The model shows a mean absolute error (MAE) of 1.3% and 1.8% in terms of maximum displacement of the inversion layer and maximum pressure disturbance, respectively. Moreover, the model underestimates the velocity over the wind-farm area with a MAE of 5.6%. Note that the three-layer model is a linearized model, hence the discrepancies with LES results increase with increasing perturbation values. In fact, the model agrees very well with LES data when perturbations are small (i.e, when non-linear effects are negligible). For further details, we refer to Allaerts and Meyers (2019).*"

Before to apply the results obtained to a higher validity model (i.e, our in-house LES solver SP-Wind), we need to improve the LES setup, which is what we will do in the near future. However, we did not mention it in the article, therefore we have added the following sentence to the last paragraph of the conclusions (P22-L531):

"*In the future, we also plan to apply the results obtained in this article to a higher fidelity model (i.e, our in-house LES solver SP-Wind). However, this requires some work on the efficiency of non-reflecting boundary conditions in our LES solver (Allaerts and Meyers 2017, 2018).*"

Finally, to avoid mentioning only the highest energy gain found in the abstract , we have modified the last sentence to (P1-L16):

"*Overall, energy gains above 4% were observed for 77% of the cases with peaks up to 14% for weakly stratified atmospheres in critical flow regimes.*"

- 3) Generally, assumptions should be stated clearly. For example, the wakes between the turbines are not explicitly modelled. This is a large assumption to make, and is only briefly mentioned in the text. What is the expected impact on the results? How does it affect the conclusions drawn in the article? Also, the sampling time seems rather large for typical wind farm control algorithms. How does this impact your results? Would you be able to find a periodic optimal signal if you had a shorter sampling time? How about the fidelity of your rotor model – would things change with an ALM model?
  We agree with the referee that the sentence "the wakes between the turbines are not explicitly modelled" is misleading. To model the farm drag force, we use a box-function wind-farm force model (also used in Smith (2010) and Allaerts and Meyers (2019)) which uniformly spreads the force over the simulation cells in the wind-farm area and does not represent the disturbances caused by each turbine in detail. The force magnitude depends on the wind-farm layout (see parameter $\beta$), the wind speed, and the thrust-coefficient distribution (i.e., the $C_T$ value in every grid cell within the farm). To avoid confusion, we have modified the sentence to (P4-L111):

"*We use a box-function wind-farm force model similar to Smith (2010) in our study. This al-*

*lows us to avoid the complexity of wake models while gaining in computational time. In fact, this model uniformly spreads the force over the simulation cells in the wind-farm area and does not represent the disturbances caused by each turbine in detail. The force magnitude depends on the wind-farm layout, the wind speed and the thrust set-point distribution (i.e., the $C_T$ value in every grid cell within the farm).*"

To understand how this simple wind-farm model affects the conclusion drawn in the article, it is useful to compare the three-layer model predictions using both the model previously discussed and the Gaussian wake model, which are shown in Allaerts and Meyers (2019) (VAL2 and VAL3, respectively). Although the magnitude of the predictions are slightly different, the trends are unchanged. Hence, we expect that trends may remain the same if a more accurate force model would be used. Nevertheless, the accuracy of the results could benefit from an improved force model (this also answer to the question regarding the ALM). Future work needs to focus on further improving the model, as well as on validation.

Regarding the question on the sampling time, gravity waves have a different time scale than wake convection, which justifies the larger sampling time used. We have added this consideration in Section 4.1 (see second comment). We have also tried to use smaller sampling time (i.e, down to a couple of seconds) but the optimizer has never found an unsteady optimum.

- 4) The article is long, making it cumbersome to read. Perhaps certain parts can be omitted. For example, is the model from section 2.1 a novel contribution or is it identical to the one described in Allaerts and Meyers 2019? If the latter, consider removing it from this article. The model described in section 2.1 is similar to the one discussed in Allaerts and Meyers (2019), but nevertheless, we have added the time dependency to the equations, a different equations' form is used, and the wind-farm force model is different. Therefore, we believe that a brief explanation of the model equations makes the article more understandable (the model description occupies approximately a page and a half, excluding the wind-farm force model). This is the reason why we decided to include this section in the article. However, to reduce the length of the section (and of the article in general), we have deleted some unnecessary sentences and explanations from the text.

- 5) Figure 1: It seems as if you have very few iterations before convergence. Can you comment on this? Figure 1 shows that the cost function decreases rapidly in the firsts two to three iterations, reaching convergence after approximately 5 algorithm iterations. The use of a quasi-Newton method in combination with the limited complexity of our optimization model (for instance, the constraints are linearized equations) allow us to reach such a fast convergence (e.g., note that a Newton method reaches convergence in one step for a classical convex QP, i.e. convex quadratic cost function with linear constraints). Moreover, the continuous adjoint method limits the number of function evaluations, since it is not necessary to evaluate $\tilde{\mathcal{J}}(C_T + \alpha \delta C_T)$ for all directions $\delta C_T$ in the control space (at the expenses of solving an auxiliary set of equations). Based on this, we were not surprised in reaching convergence after 6 L-BFGS iterations with only 20 function evaluations. To include these considerations in the article, we have added the following sentence in section 3.1 (P9-L246):

"*Fig. 1 shows that the cost function decreases rapidly in the firsts two to three algorithm iterations, reaching a plateau afterwards. The use of a quasi-Newton method in combination with the limited complexity of our optimization model (for instance, the constraints are*

*linearized equations) allow us to reach such a fast convergence. Moreover, the continuous adjoint method limits the number of function evaluations, since it is not necessary to evaluate $\tilde{\mathcal{J}}(C_T + \alpha \delta C_T)$ for all directions $\delta C_T$ in the control space (at the expenses of solving an auxiliary set of equations).*"

---

## Author Comment (AC3) · 27 Aug 2020

**Response to comments of Alan Wai Hou Lio**

- The paper is interesting. The authors investigated the problems of the mesoscale interaction between a wind farm and atmospheric boundary-layer. A three-layer model is proposed for modelling the wind-farm induced gravity wave. Based on the simplified model, optimisation is then proposed to find the optimal thrust coefficient distributions that maximise the wind farm power output. The concept is appealing and the topic is definitely relevant to the wind energy community.

  We would like to thank the referee for the constructive feedback in improving the quality of the paper.

- Comments: pg1: Some claims by the authors were not clear. For example, the optimal thrust coefficient distributions are spatially stationary rather than time-periodic. Did the authors consider turbulent wind inflow and turbine-to-turbine interactions? Does the claim imply that stationary spatial distributions of thrust coefficients are better than dynamically changing the thrust set-points for maximising the wind farm power? This claim disagreed with some of the other works (e.g. [1]). In [1], the benefit of periodic dynamic induction control was shown, where the thrust coefficient of the upstream turbine was periodically adjusted to improve the downstream wind flow. How is this work related to [1]?

  The referee is right in saying that our claims disagree with some previous works. Although our study deals with much larger scales than the ones usually considered in dynamic or static induction control, we believe that it is useful to relate it with other recent findings. Hence, the following paragraph has been modified at P12-L307:

  *"The optimization model described in Section 2.2 is time and space dependent. Hence, the model is capable of finding a time-periodic optimal thrust-coefficient distribution over the wind-farm area in a fixed time interval $[0, T]$. However, all optimal thrust set-point distributions found for the different combinations of time horizons and time steps reported in Table 1, are constant in time. We have verified this using a range of steady and unsteady starting conditions for $C_T$ in the algorithm, but did not find any unsteady optimum. We believe that this is due to two reasons. Firstly, we use steady-state inflow conditions, therefore neglecting meso-scale temporal variations in the velocity field (these could lead to time-dependent optimal control signals, but are not included in the current work). Secondly, the objective function is non-convex and there is no proof about the uniqueness of global minima. Hence, there is no guarantee that the optimal solution found by the optimizer corresponds to a global optimum. Nevertheless, since we do not observe any unsteady behaviour in our optimal solutions, we show only steady-state results in the remainder of the manuscript, and conclude for the time being that unsteady time-periodic excitation is less effective than a stationary spatially optimal distribution in this context.*

  *We also note that our findings are in contrast with recent works of Goit and Meyers (2015), Munters and Meyers (2018) and Frederik et al. (2020), in which the authors illustrated the benefits of dynamic induction control over yaw and static induction control. However, the characteristic time scale of gravity-wave effects is estimated to be approximately 1 h (Gill 1982, Allaerts and Meyers 2019) which is an order of magnitude above the typical time scale of wake convection between turbines, and turbulent mixing in turbine wakes (this also justifies the larger sampling time used). Hence, while unsteadiness of the thrust coefficient (with a typical time scale of 50 seconds for large scale turbines) can lead to improved wake mixing (Goit and Meyers (2015), Munters and Meyers (2018), Frederik et al (2020)), it has no impact on phenomena that occur at larger time scales, such as wind-farm induced gravity waves."*

- pg2 l59: "asses" -> asks.
  Thank you, we have corrected this erratum.

- pg4 l116: Ct is a function of Ct(x,y,t)? What is x and y in B(x,y)?
  To avoid the influence of the wind-farm layout on the results presented, the wind profile is always oriented along the x-axis. Hence, the x- and y-axis denote the streamwise and spanwise direction, respectively. The function $B(x,y)$ is a box function equal to one for the $(x,y)$ coordinates within the wind-farm area and zero elsewhere. Similarly, the thrust-coefficient distribution $C_{\mathrm{T}}(x,y,t)$ is function of the spatial coordinate x and y and of the time t. The thrust- and power-coefficient distribution are always multiplied by the box function $B(x,y)$ in the text, since they are defined only within the wind-farm area. The following sentence has been added at P5-L122:

  "*the x- and y-axis denote the streamwise and spanwise direction, respectively.*"

- pg5 l123: What is the dimension of Ct? Is Ct a vector where the number of elements in that vector is equal to the number of turbines? How is Ct of each turbine related the aggregate wind farm drag f?
  $C_T = C_T(x,y,t)$ denotes the thrust set-point distribution and can be represented mathematically as $C_T : \mathbb{R}^2 \times [0,T] \to \mathbb{R}$. In the text, the thrust-coefficient distribution is always multiplied by the box function $B(x,y)$ (see previous comment) so that it assumes non-zero values only within the wind-farm area. If we denote with $N_x^{\mathrm{wf}}$ and $N_y^{\mathrm{wf}}$ the number of grid points within the farm along the x- and y-direction, $N_x^{\mathrm{wf}} N_y^{\mathrm{wf}}$ represents the number of grid cells in the farm area. We assume a constant $C_T$ value in every cell, and each of these values represent a control parameter of our optimization problem. Hence, the number of control parameters is given by $N_x^{\mathrm{wf}} N_y^{\mathrm{wf}} N_t$ (this is mentioned in the article at P7-L190).

  In regards to the last question, note that we select the turbine spacings $s_x$ and $s_y$ to have a density of turbines in the farm similar to the one of Allaerts and Meyers (2019) (i.e. leading to $\beta = 0.01$ in Eq. 8, using $\eta_w = 0.9$ and $\gamma = 0.9$ similar to Allaerts and Meyers (2018)). Hence, we fix the density of turbine in the farm but we do not specifically define a layout or a number of turbines. In fact, the force model uniformly spreads the force over the simulation cells in the wind-farm area and the number of grid cells within the farm define the DOF of our optimization problem (as mentioned above). The turbine spacings (together with other parameters) only define the wind-farm drag-force magnitude. In order to compute the thrust-coefficient $\tilde{C}_{T,k}(t)$ of a turbine at location $(x_k, y_k)$, it is possible to evaluate the thrust coefficient distribution $C_T(x_k, y_k, t)$. A more accurate connection between $\tilde{C}_{T,k}(t)$ and the drag force $f$ would require the use of an analytical wind-farm model, but this is out of the scope of the current work. To include these information in the text, we have added the following sentence at P5-L127:

  "*Finally, $C_T(x,y,t)$ represents the thrust-coefficient distribution. To compute the thrust coefficient $\tilde{C}_{T,k}(t)$ of a turbine at location $(x_k, y_k)$, it is possible to evaluate the thrust set-point distribution $C_T(x_k, y_k, t)$. A more accurate connection between $\tilde{C}_{T,k}(t)$ and the drag force $f$ would require the use of an analytical wind-farm model, but this is out of the scope of the current work.*"

  Moreover, we have also mentioned at P10-L267 that:

  "*The wind turbine relative spacings along the x- and y-direction are $s_x = s_y = 5.61$ (both non-dimensionalized with respect to the turbine rotor diameter $D$), so that the density of*"

*turbines in the farm is similar to the one of Allaerts and Meyers (2019) (i.e. leading to $\beta = 0.01$ in Eq. 8, setting both the wake efficiency $\eta_w$ and $\gamma$ to 0.9 as in Allaerts and Meyers (2018)). Note that we do not define a specific layout or a number of turbines but we only fix the density of turbines in the farm."*

- p5 l124: "the thrust-coefficient distribution Ct has to be interpreted as a perturbation." Is Ct the thrust coefficient or the perturbation to the thrust coefficient?
The equations are linearized with respect to the background state variables, that is around a state for which the wind-farm is not operating. Hence, there is no distinction between the thrust coefficient $C_\mathrm{T}$ and its perturbation $C_\mathrm{T}'$ . In fact, $C_\mathrm{T} = C_\mathrm{T}^\mathrm{b} + C_\mathrm{T}'$ but we are linearizing around a non-operating wind farm, therefore $C_\mathrm{T}^\mathrm{b} = 0$ and $C_\mathrm{T} = C_\mathrm{T}'$. However, in an attempt to reduce the length of the article (suggested by the Anonymous Referee #2), we have decided to remove this sentence from the text (a similar explanation is already written in Allaerts and Meyers (2019)).

- "The goal of the optimization framework is to find a time-periodic optimal thrust-coefficient distribution". Why did the authors assume that the optimal thrust distribution would be time-periodic in the beginning?
We are using steady environmental conditions. Hence, if we use a control signal for a finite time window $[0, T]$, since nothing changes in the atmospheric conditions, the only option is that the signal repeats itself (at least if we want to arrive at a control that is on average steady) in the time window $[T, 2T]$, and so on. To include this information in the article, we have modified the text as follows (P6-L163):

*"The goal of the optimization framework is to find a time-periodic optimal thrust-coefficient distribution $C_T^\mathrm{O}(x, y, t)$ that minimizes the gravity-wave induced blockage effects, maximizing the flow wind speed and consequently the wind-farm energy extraction over a selected time period $T$. The background atmospheric state is presumed to be steady, which is the reason why we use a time-periodic control (i.e. leading to a moving time average of the optimal control that is steady, and does not lead to end-of-time effects)."*

- p6: Equation (13), what is $\psi$ and J in (13) is not a function of Ct. I suggest the authors swap equation (13) and (14) for clarity.
$\psi$ is the vector containing the state variables and is defined in P7-L189.
Correction (P6-L169):

*"We have swapped equation (13) with equation (14) as suggested by the referee. Hopefully this will help in clarifying the dependence of J on $C_\mathrm{T}$."*

- p7 l183: $\mathcal{N}(\boldsymbol{\psi}(C_t), C_t) = 0$. Is this only valid around the neighbourhood of the solution? What is $\mathcal{N}$?
The vector $\boldsymbol{\psi}$ contains the state variables, hence it depends upon the thrust coefficient distribution. We denote with $\boldsymbol{\psi}(C_T)$ the solution of the state equation. $\mathcal{N}(\boldsymbol{\psi}, C_T)$ is an operator which represents the state equations. If $\boldsymbol{\psi}(C_T)$ is a solution, then $\mathcal{N}(\boldsymbol{\psi}(C_T), C_T)$=0. This does not hold in the neighbourhood of the solution but only for $\boldsymbol{\psi}(C_T)$ which satisfies the state equations. The sentence has been changed to (P7-L195):

*"To avoid exploring the entire feasibility region, we require $\boldsymbol{\psi}(C_T)$ to be the solution of the state equations throughout the optimization process. In other words, defining an operator $\mathcal{N}(\boldsymbol{\psi}, C_T)$ that denotes the state equations, we are enforcing $\mathcal{N}(\boldsymbol{\psi}(C_T), C_T)= 0$ during optimization iterations."*

- p11 l268: what is $P_N$?
  The non-dimensional number $P_N$ is defined as $P_N = U_B^2/NH\|\boldsymbol{U}_g\|$ where $N$ is the Brunt-Väisälä frequency. This number is an indicator of the effects of internal waves in the troposphere. For instance, low $P_N$ values correspond to strongly stratified atmosphere which in turn implies strong excitation of internal waves. The following sentence has been added (P12-L298):

  "*Further, $P_N$ expresses the impact of internal waves in the troposphere which increases when $P_N$ decreases. The background state defined in Table 1 leads to $P_N = 1.92$.*"

---

## Editor Decision (ED1)

General comments
- Very interesing study but could be improved quite a bit through a stronger introduction that helps motivate and contextualize the issue further.
- A major weakness seems to be the overall set up and architecture of the optimization (the problem formulation). One issue is the ad hoc case study selection. That can be reasoned away to an extent and the authors have sought to do so- not as well as I would like but well enough. More importantly, though, if I understand correctly, inherent in the optimization of the thrust coefficients are flow effects both having to do with the atmospheric effects and the wake effects. Thus, there are multiple physical phenomena driving the results and these need to be disentangled at least in explanation if not in the analysis itself. I may be mistaken on this, in which case please clarify further in the paper on this front.

Additional detailed comments by section:

Introduction
- Consider a more fundamental definition and description of gravity waves for those who are not familiar with the concept. A more general description and then the concept particularly applied to wind energy induced phenomena
- Thrust coefficient manipulation is an intermediate effect that is brought about by wind farm control. The distinction should be addressed
- Generally, the introduction seems to jump into details without enough context
- Lines 44-48 – the way it is written, the concepts of blockage and gravity waves are being confounded
- Literature review on wind farm control is weak. There is a lot more work in the space including the comprehensive review article from 2019
- Wind farm layout role in production is weak – there is a vast literature on optimization around wind turbine spacing considering multidisciplinary concerns with AEP a very large subset of said literature. I'm not even sure why this topic is thrown in here unless layout optimization is a consideration in this paper
- The discussion in the introduction of the methodology proposed doesn't address any validation – this is done in section 2 but could be done here as well.
- In general, the introduction feels a bit incomplete – insufficiently motivated, insufficient description of concepts and insufficient discussion of prior art and how this work uniquely extend from it

Methodology
- For the validation of the three-layer model, the discussion in lines 155-161 seems quite limited. Is there anything more that can be said about the reasons for the error and underestimation of velocity beyond generic model fidelity arguments? How will these errors be expected to affect the current optimization study?
- At the beginning of section 2.2, consider adding a general discussion of how the thrust affects induction and interaction with gravity-wave blockage… this could also be brought forth in the introduction

Numerical setup and case description

- The description of the computational costs is loose and could be stronger and tabularized in terms of function evaluations, etc
- Typo line 246 firsts
- The choice of wind-farm layout / case is not well justified. Generally, it would be good to see a two-fold approach where a smaller illustrative case is used to explore the effects of the drivers on the optimization problem and then the application in a larger case study. The ad hoc nature of using a large case leaves in question the generalizability of the results
    o The handling of the atmospheric states seems more in line with an approach to explore drivers under different conditions

Results and discussion
- Line 314 – uniqueness is wrong word. You are not guaranteed with your optimization approaches of finding a global optima for a nonconvex problem such as this. That is certainly true. However this seems to be an odd argument for rationalizing the fact that there is not an unsteady optimum…
    o The latter point about time-scales seems much more relevant. I recommend striking the entire local/global discussion at least in this context
- Figure 3 seems a bit disconnected. Fig 2 was nice but it would be nice to show some sort of relative effect on the inversion-layer displacement after the optimal Ct setpoints are found.
- Section 4.1 could be strengthened by a summary table of key statistics for each of the cases…
- The language around the resulting optima is strange. You discuss sinusoidal behavior of the setpoints which is an odd way of saying that there is periodic pattern in subsequent rows of turbines in terms of the optimal setpoints. Try to tie this back more to the reality of what is going on with the turbines. These aren't mathematical features in a CFD world, these are turbines in a farm. Each turbine is a unique entity with a vector of design variables for its Ct setpoint over time
    o Honestly, I don't get why you would have a spatially invariant Ct as a sensitivity study… that makes no sense to me at all. In practice you would never try to force uniformity of Ct. Make sure what you do makes sense in reality even if you have to abstract and simplify away from it.
- I find the explanation of the results in section 4.1 generally weak.  Can you tie things more to the physics at play? Maximizing for energy will drive your optimal set points to a certain setting already to mitigate wake effects. The atmospheric effects are another layer. Is there any coupling? Did you do the optimization without the gravity waves and optimize the setpoints of the thrust first? This would be good to do in order to investigate the influence of each of the phenomena separately. Optimizing the thrust without disentangling the two means that you may be attributing too much of the effect to the counteraction of the influence of the atmospheric state
- I understand you are reporting energy gains because you are time integrating power. But still, these are gains for a particular inflow condition set… so the energy gains reported (particularly in the abstract) could be easily misinterpreted… energy gain in the world of wind farm optimization (for control or other) typically looks from an annual perspective. Gains for particular inflow condtions are generally reported as power gains

Sensitivity study
- Again, mentioning the wind farm layout is out of scope is odd. I think it goes back to the architecture of the study where a case study was selected ad hoc rather than building up from a set of canonical cases. It would be nice to see a follow on conference article go back and do a more exhaustive exploration. It is not clear to me why the layout (at least the spacing of turbines) would not be a key sensitivity done in the current study… to me, that is indeed a key sensitivity
- The study is interesting, but it could be made more accessible through better context. How often do these different conditions happen in reality?

Appendices
- Recommend deleting appendix B – see prior notes

---

## Author Response (AR2)

**1  Response to comments of Katherine Dykes**

- Very interesing study but could be improved quite a bit through a stronger introduction that helps motivate and contextualize the issue further. A major weakness seems to be the overall set up and architecture of the optimization (the problem formulation). One issue is the ad hoc case study selection. That can be reasoned away to an extent and the authors have sought to do so- not as well as I would like but well enough. More importantly, though, if I understand correctly, inherent in the optimization of the thrust coefficients are flow effects both having to do with the atmospheric effects and the wake effects. Thus, there are multiple physical phenomena driving the results and these need to be disentangled at least in explanation if not in the analysis itself. I may be mistaken on this, in which case please clarify further in the paper on this front.
  We would like to thank the editor for the very constructive feedback. We will address the questions and requests below.

**Introduction**

- Consider a more fundamental definition and description of gravity waves for those who are not familiar with the concept. A more general description and then the concept particularly applied to wind energy induced phenomena
  To provide a better definition and description of gravity waves, we have added the following paragraph at P1-L24:

  "In a stable atmosphere, an air parcel which is vertically perturbed will have the tendency to fall back to its original position. In such case, an oscillation is initiated that is driven by gravity and inertia; this is called a gravity wave. Mountains are examples of orographic obstacles that trigger vertical flow displacement, and consequently gravity waves (Smith, 1980). The drag force exerted by the mountain is usually transported upward by these waves. At the point of breakdown, the drag force is released in the upper levels of the atmosphere, causing a slow down of the large-scale flow (Eliassen and Palm 1960; Durran, 1990). Moreover, when air is lifted in a stable atmosphere, a cold anomaly is created, which induces horizontal pressure gradients (Smith, 2010).
  In a wind farm, the upward displacement of the boundary layer, caused by diverging fluid streamlines due to flow deceleration by the turbines, can trigger gravity waves in the stable free atmosphere above the boundary layer as well (Smith, 2010; Allaerts and Meyers, 2017). As a result, an adverse pressure gradient develops in the induction region of the wind farm, which slows down the wind-farm inflow velocity (Allaerts and Meyers, 2018). The size of this region scales with the length of the farm. This phenomenon is one possible cause of flow blockage. Note that it differs from classical hydrodynamic blockage caused by the turbine induction, which typically scales with the turbine rotor diameter, and which has also been studied recently in much detail (Bleeg et al., 2018; Segalini and Dahlberg, 2019). "

- Thrust coefficient manipulation is an intermediate effect that is brought about by wind farm control. The distinction should be addressed
  To clarify our approach, we have added the following sentences at P3-L72:

  "Note that we do not use the tip-speed ratio and/or the pitch angle distribution as control parameters. Instead, we directly control the thrust set-point distribution. In fact, the former approach would not add further insight in the study performed in the current manuscript."

- Generally, the introduction seems to jump into details without enough context
  To add more context in the introduction, we have included a more fundamental description of gravity waves and a more consistent literature review about wind-farm control (see also below).

- Lines 44-48 – the way it is written, the concepts of blockage and gravity waves are being confounded
  Gravity waves are one possible cause of flow blockage in the induction region of a wind farm. In order to clarify this, we have added the following sentences at P2-L33:

  "As a result, an adverse pressure gradient develops in the induction region of the wind farm, which slows down the wind-farm inflow velocity (Allaerts and Meyers, 2018). The size of this region scales with the length of the farm. This phenomenon is one possible cause of flow blockage. Note that it differs from classical hydrodynamic blockage caused by the turbine induction, which typically scales with the turbine rotor diameter, and which has also been studied recently in much detail (Bleeg et al., 2018; Segalini and Dahlberg, 2019)."

- Literature review on wind farm control is weak. There is a lot more work in the space including the comprehensive review article from 2019
  We agree with the statement made by the editor. Hence, we have written differently this paragraph. You can find the revised version here below or at P3-L61:

  "In the last decades, a considerable amount of research has focused on wind-farm control strategies that allow to maximize the farm power output. We refer to Kheirabadi and Nagamune (2019) for a recent comprehensive overview. However, earlier studies all focus on influencing wake dynamics and wake mixing, which occur at a much smaller scale than wind-farm induced gravity waves, to improve power extraction in waked turbines. Important control mechanisms include wake redirection (by yawing and tilting of the turbine), and turbine de-rating strategies. Control actions that influence wind-farm physics on a much large scale, such as self-induced gravity waves, are not explored to date."

- Wind farm layout role in production is weak – there is a vast literature on optimization around wind turbine spacing considering multidisciplinary concerns with AEP a very large subset of said literature. I'm not even sure why this topic is thrown in here unless layout optimization is a consideration in this paper
  Layout optimization is not considered in the current article. Therefore, we have removed from the text the following sentences:

  ""

- The discussion in the introduction of the methodology proposed doesn't address any validation – this is done in section 2 but could be done here as well.
  We presume that the editor is suggesting to add comments about validation in the introduction of section 2 (rather than at the end of the section). This is a good suggestion. Consequently, we have moved up the discussion about model validation (P4-L98). Also, as suggested, we have extended this discussion (see comment below).

- In general, the introduction feels a bit incomplete – insufficiently motivated, insufficient description of concepts and insufficient discussion of prior art and how this work uniquely extend from it

  To improve the readability of this section, and consequently of the overall manuscript, we have included a more fundamental description of gravity waves. Next, we have modified the text so that the concept of flow blockage is not confounded with gravity waves. Moreover, we have written differently the paragraph about wind-farm control. Finally, to explain how our work differ from others in literature, we have added the following sentences at P3-L65:

  "... Control actions that influence wind-farm physics on a much large scale, such as self-induced gravity waves, are not explored to date. In the current work, we concentrate on using wind-farm control to alter/improve the interaction between the wind farm and its self-induced gravity wave system. To this end, ..."

**Methodology**

- For the validation of the three-layer model, the discussion in lines 155-161 seems quite limited. Is there anything more that can be said about the reasons for the error and underestimation of velocity beyond generic model fidelity arguments? How will these errors be expected to affect the current optimization study?

  We understand the concern of the editor. In general, the underestimation in our model is a result from the linearization. The smaller the perturbation, the better the model can potentially fit reality, but this will require further experimental validation in the future. From this perspective, it may be expected that errors decrease slightly in optimized settings in which displacement magnitudes are typically lower. In the manuscript, we have modified the following paragraph at P4-L98:

  "The three-layer model has been validated against LES results by Allaerts and Meyers (2019) (see Section 3 VAL2) on a two dimensional (x-z) domain (i.e., all spanwise derivatives are set to zero). The model shows a mean absolute error (MAE) of 1.3% and 1.8% in terms of maximum displacement of the inversion layer and maximum pressure disturbance, respectively. Moreover, the model underestimates the velocity over the wind-farm area with a MAE of 5.6%. Note that the three-layer model is a linearized model, hence the discrepancies with LES results increase with increasing perturbation values. In fact, the model agrees very well with LES data when perturbations are small (i.e, when non-linear effects are negligible). From this perspective, it may be expected that errors decrease slightly in optimized settings in which perturbation magnitudes are typically lower. For further details about model validation, we refer to Allaerts and Meyers (2019)."

- At the beginning of section 2.2, consider adding a general discussion of how the thrust affects induction and interaction with gravity-wave blockage... this could also be brought forth in the introduction

  Thank you for the suggestion. We have added the following discussion at the end of the first paragraph of section 2.2 (P6-L182):

  "Note that the relation between overall wind-farm drag and wind-farm blockage is non-trivial. On the one hand, increased wind-farm drag leads to increased wind-farm blockage induced by gravity waves. This results from mass conservation and the upward displacement of the free atmosphere. On the other hand, increased wind-farm blockage reduces wind-farm drag. Thus, the aim of the optimization is to find the optimal balance between these two opposing trends."

**Numerical setup and case description**

- The description of the computational costs is loose and could be stronger and tabularized in terms of function evaluations, etc

  In the article, we only give an idea of the time required to compute the optimum. This was our aim. Looking back, we could have run the optimization algorithm with different initial conditions and different input parameters and further tabularize the function evaluation cost and the total simulation time. However, this work was done a year ago and, in the meantime, we have speeded-up the equation solver of several order of magnitude (as mentioned in the text). Therefore, we have decided to not add further information on this aspect. In a future work, we will make sure to provide a more detailed computational cost analysis.

- Typo line 246 firsts

  Thank you. We have corrected the erratum.

- The choice of wind-farm layout / case is not well justified. Generally, it would be good to see a two-fold approach where a smaller illustrative case is used to explore the effects of the drivers on the optimization problem and then the application in a larger case study. The ad hoc nature of using a large case leaves in question the generalizability of the results (the handling of the atmospheric states seems more in line with an approach to explore drivers under different conditions)

  To better justify the choice of the wind-farm layout, we have added the following sentences at P10-L281:

  "Allaerts and Meyers (2019) conducted a sensitivity study on the effects of wind-farm layout on gravity-wave induced power losses. They show that these power losses monotonically increase with the size of the farm. Also, they mention that the losses are maximum when the wind-farm ratio $L_y/L_x$ is close to $3/2$, while being negligible for very wide but short farm, and vice versa (i.e., negligible for $L_y/L_x \ll 1$ and $L_x/L_y \gg 1$). Since we are interested in optimal thrust coefficient distributions in presence of gravity waves, we have selected the "worst-case" wind-farm layout (i.e., a wind-farm width and length of $L_y = 30$ km and $L_x = 20$ km, respectively). We note that this was also the farm layout chosen by Allaerts and Meyers (2018, 2019), which resembles in size the Belgian-Dutch wind-farm offshore cluster located in the North Sea, but simplified to a rectangular shaped wind-farm. Also, Smith (2010), Fitch et al. (2012) and Wu and Porté-Agel (2017) have used a farm with similar dimensions in their studies."

**Results and discussion**

- Line 314 – uniqueness is wrong word. You are not guaranteed with your optimization approaches of finding a global optima for a nonconvex problem such as this. That is certainly true. However this seems to be an odd argument for rationalizing the fact that there is not an unsteady optimum. . . (the latter point about time-scales seems much more relevant. I recommend striking the entire local/global discussion at least in this context)

  We have tried to give three different reasoning to the fact that we do not observe any unsteady behaviour in our optimal solutions. However, as recommended, we have deleted the local/global discussion, i.e., the following sentences:

  "

Table 2: Relative change in percentage between optimal and reference maximum flow perturbation values. Power gains are also included.

|  | $\mathbf{F_r = 0.9}$ | $\mathbf{F_r = 1.1}$ |
|---|---|---|
| Maximum inversion-layer displacement | $-14.5\%$ | $-16.8\%$ |
| Maximum pressure perturbation | $-14.3\%$ | $-16.2\%$ |
| Maximum velocity perturbation | $-13.4\%$ | $-15.5\%$ |
| Power gain | $5.3\%$ | $7.0\%$ |

"

- Figure 3 seems a bit disconnected. Fig 2 was nice but it would be nice to show some sort of relative effect on the inversion-layer displacement after the optimal Ct setpoints are found.
  We have organized section 4.1 as follows. First, in Fig. 2, we show the three-layer model predictions using the reference thrust coefficient distribution (i.e. $C_\mathrm{T}^\mathrm{R}(x,y) = C_\mathrm{T}^\mathrm{Betz} = 8/9$). Then, the optimal thrust coefficient distributions for both sub- and super-critical flow are illustrated in Fig. 3. Finally, in Fig. 4 and 5, we compare the inversion-layer displacement, pressure and velocity perturbations computed with the three-layer model using the reference and optimal $C_\mathrm{T}$ distributions. These effectively contain the comparisons that the editor is asking for. In these figures, we show the results as line plots along the center line of the wind farm. We do not show planforms of the inversion-layer displacement etc, since we observed that this type of plot does not visualize the differences very well.

- Section 4.1 could be strengthened by a summary table of key statistics for each of the cases. . .
  We agree with the editor's statement. Hence, we have added a table in section 4.1, which summarizes some key statistics. For simplicity, we have reported the table here– see Table 2.

- The language around the resulting optima is strange. You discuss sinusoidal behavior of the setpoints which is an odd way of saying that there is periodic pattern in subsequent rows of turbines in terms of the optimal setpoints. Try to tie this back more to the reality of what is going on with the turbines. These aren't mathematical features in a CFD world, these are turbines in a farm. Each turbine is a unique entity with a vector of design variables for its Ct setpoint over time
  We understand the remarks of the editor. However, as mentioned in section 2.1, the farm drag model used does not represent turbines explicitly. Therefore, we cannot rephrase the sentences as suggested. To give this kind of interpretation to the results, we would, e.g., need to couple an analytical wind-farm wake model to the gravity wave model. Then, each turbine would be a unique identity with a vector of design variables (e.g., $D$, $z_\mathrm{hub}$, $C_\mathrm{T}$, etc..) associated. In the manuscript, we tried to better explain this at P5-L145:

  "A more accurate connection between $\widetilde{C}_{T,k}(t)$ and the drag force $\boldsymbol{f}$ would, e.g., require the use of an analytical wind-farm wake model. This is however not considered in the current work, so that wake effects are not explicitly incorporated in the optimization. Rather, we consider the optimization of the gravity-wave system, while presuming that the wake

efficiency parameter $\eta_{\mathrm{w}}$ does not change as a result of the optimization."

Finally, note that representing turbines as a smoothed thrust distribution that is spread out throughout the farm is, e.g, quite common in regional climate models (in which multiple turbines can fall in a computational cell). A similar force model has also been used before in gravity-wave studies by Allaerts and Meyers (2018, 2019) and in Smith (2010)).

- Honestly, I don't get why you would have a spatially invariant Ct as a sensitivity study... that makes no sense to me at all. In practice you would never try to force uniformity of Ct. Make sure what you do makes sense in reality even if you have to abstract and simplify away from it.
  In the first manuscript version uploaded on WES, we did not consider to optimize for a uniform set-point distribution. However, in the first round of review, Dries Allaerts asked us the following questions:

  Did you consider optimizing for a uniform set-point distribution? What are the maximum gains to be expected there, and hence how much more is there to be gained by using a non-uniform set-point distribution? How would that uniform value compare to the average of the non-uniform distribution, and would the uniform value depend on the atmospheric condition as well?

  Hence, Appendix B has been added to address these questions. We understand that forcing uniformity in $C_{\mathrm{T}}$ is not something that you would do in practice. However, in this framework, it allows us to quantify the gains that come from an optimal non-uniform distribution. Moreover, under the assumption of uniform $C_{\mathrm{T}}$ distribution, the control space has unitary dimension. Therefore, the adjoint equations are not needed to solve the optimization problem and the model find the optimum in a much shorter time. Despite this, the power gains found are still considerable.
  Since this analysis was explicitly asked by a reviewer, we prefer to not remove appendix B from the text - in the end, it remains an appendix, and can be easily skipped when reading the paper.

- I find the explanation of the results in section 4.1 generally weak. Can you tie things more to the physics at play? Maximizing for energy will drive your optimal set points to a certain setting already to mitigate wake effects. The atmospheric effects are another layer. Is there any coupling? Did you do the optimization without the gravity waves and optimize the setpoints of the thrust first? This would be good to do in order to investigate the influence of each of the phenomena separately. Optimizing the thrust without disentangling the two means that you may be attributing too much of the effect to the counteraction of the influence of the atmospheric state
  Our model does not incorporate wake losses explicitly, except by means of the wake efficiency parameter $\eta_{\mathrm{w}}$ that we presume constant during optimization. Thus, our optimization is performed under the assumption that wake losses do not change, and only gravity waves are affected. In the future, it will be interesting to see whether including wake losses can increase or rather reduce the potential for overall power gains in the wind-farm operation.
  In the manuscript, we better discussed this in section 2.1 (see comment above), and further we also modified the following paragraph at P23-L550:

  "... before this can be translated to real wind-farm applications. In the current work, we did not include an explicit wake model in our model, and have presumed that wake losses remain unchanged during optimization (i.e., $\eta_{\mathrm{w}}$ is assumed to be constant). In the

future, an analytical wind-farm wake model, such as, e.g., the one developed by Niayifar and Porté-Agel (2016) and used by Allaerts and Meyers (2019) could be adopted for optimization. This would however also require better representation in the wake model of changing background variables and pressure gradients. For instance, gravity-wave induced pressure gradient effects on turbine wake recovery could be included using the model proposed by Shamsoddin and Porté-Agel (2018) that incorporates effects of pressure gradients. Furthermore, the use of a wind-farm drag model which computes analytically the wake of each turbine would allow us to investigate separately the influence of wake effects and gravity waves on the optimal turbine set-points. This is work for future research."

- I understand you are reporting energy gains because you are time integrating power. But still, these are gains for a particular inflow condition set... so the energy gains reported (particularly in the abstract) could be easily misinterpreted... energy gain in the world of wind farm optimization (for control or other) typically looks from an annual perspective. Gains for particular inflow conditions are generally reported as power gains

  We agree with the editor's statement. Therefore, to avoid confusion with annual energy production studies, we have written all gains in terms of power instead of energy. Also, we have changed the definition of gain as follows:

  "We denote with $\mathcal{P}^{\mathrm{R}} = \widetilde{\mathcal{J}}^{\mathrm{R}}/T$ and $\mathcal{P}^{\mathrm{O}} = \widetilde{\mathcal{J}}^{\mathrm{O}}/T$ the power extracted using $C_T = C_T^{\mathrm{R}} = 8/9$ and $C_T = C_T^{\mathrm{O}}$, respectively. Further, we define

  $$\mathcal{G} = \frac{\mathcal{P}^{\mathrm{O}} - \mathcal{P}^{\mathrm{R}}}{\mathcal{P}^{\mathrm{R}}}$$

  where $\mathcal{G}$ denotes the relative power gain obtained using an optimal thrust-coefficient distribution instead of the reference one. Note that the optimal distributions are steady-state, therefore the power gain definition is not dependent on the choice of the time horizon $T$."

**Sensitivity study**

- Again, mentioning the wind farm layout is out of scope is odd. I think it goes back to the architecture of the study where a case study was selected ad hoc rather than building up from a set of canonical cases. It would be nice to see a follow on conference article go back and do a more exhaustive exploration. It is not clear to me why the layout (at least the spacing of turbines) would not be a key sensitivity done in the current study... to me, that is indeed a key sensitivity

  We mention in the article that gravity-wave induced flow blockage is also related to the farm shape and size. Therefore, we agree with the editor in saying that the layout is an important parameter to be investigated. However, this really falls outside the scope of the current study.

  To better justify the layout choice, we have added a couple of sentences at P10-L281 (see also comment above). Moreover, as suggested, we have added in the conclusion the following statement at P23-L545:

  "We note that the gravity-wave induced power losses are also sensitive to the wind-farm layout. Optimization of layout (including, e.g., relevant techno-economical constraints) is however not considered here and can be an interesting topic for future research."

- The study is interesting, but it could be made more accessible through better context. How often do these different conditions happen in reality?

For the time being, there is no complete answer to this, and we don't want to speculate too much on this in the article. In ongoing work in our group, together with colleagues from climate modeling we are identifying parameter ranges and distributions of relevant parameter conditions by means of ERA5 reanalysis. Based on this, we find that the selected parameter ranges are relevant, but we do not yet have full mapping of frequencies in, e.g., the North-Sea region. We hope to publish work on this in the future.

The current sensitivity study is based on to the one carried out by Allaerts and Meyers (2019). However, the authors used a very wide range of $h_*$, $\overline{z_0}$, $N/f_c$ and $g'H/Au_*^2$, exploring also atmospheric states that never manifest in the real word. In our study, we limit the non-dimensional groups range for two reasons. First, to ensure $U_1 < U_r$, where $U_r$ is the turbine rated wind speed. Second, to have atmospheric states that occur in reality. Although these parameter values are representative of real case scenarios, we do not yet have access to realistic frequency maps.

**Appendices**

- Recommend deleting appendix B – see prior notes
  See comment above in results and discussion section.

**2 Response to comments of Anonymous Referee #4**

- In the title, instead of referring to just "gravity waves", perhaps refer to it as "atmospheric gravity waves" to distinguish them from cosmological gravity waves. Just a small point, feel free to leave as is.
  This is a good suggestion. We have changed the title accordingly.

- In the paragraph beginning at line 111, the model for the wind-farm force is described. It seems quite simple which is fine for this study, but I am wondering what the downsides might be to using such a simple model. For future research, if a more detailed model is used, what gains might there be? Perhaps some insight in this can be included here or in the conclusion. It is mentioned that it could be improved in the conclusion, but a little more insight would be beneficial to the reader.
  The reason why we did not adopt an analytical wind-farm model is three-fold. Firstly, we wanted to avoid the computational burden which comes with these types of model. Secondly, the derivation of the adjoint of the wind-farm drag first-order term would have been much more challenging. Thirdly, wake models as they exist now are not well suited for coupling to our gravity wave model, as they do not yet allow to incorporate varying background fields and pressure gradients, and this may significantly distort optimization results. Hence, we opted for a box-function wind-farm force model, which is, as you pointed out, a very simplified representation of a farm. However, in section 3.2 and 3.3 of Allaerts and Meyers (2019), there is a comparison of three-layer model predictions obtained using a box-function and an analytical wind-farm model. If you look at Fig. 4 of Allaerts and Meyers (2019), you could see that the green and blue lines (which represent flow perturbations) have similar trend and order of magnitude. Therefore, we would expect the power gains computed with a more detailed model to be of the same order of magnitude of the ones presented in the current manuscript. To include these considerations in the text, we have added the following sentences at P5-L151:

  "We note that Allaerts and Meyers (2019) have shown that the flow perturbations computed with this simple drag force model have similar trends and orders of magnitude as the ones computed using a drag model that relies on the more detailed analytical wake model of Niayifar and Porté-Agel (2016). Therefore, we believe that the model adopted is a reasonable representation of reality."

  Moreover, we added on P23-L550:

  "... before this can be translated to real wind-farm applications. In the current work, we did not include an explicit wake model in our model, and have presumed that wake losses remain unchanged during optimization (i.e., $\eta_w$ is assumed to be constant). In the future, an analytical wind-farm wake model, such as, e.g., developed by Niayifar and Porté-Agel (2016) and used by Allaerts and Meyers (2019) could be adopted for optimization. This would however also require better representation in the wake model of changing background variables and pressure gradients. For instance, gravity-wave induced pressure gradient effects on turbine wakes recovery could be included using the model proposed by Shamsoddin and Porté-Agel (2018) that incorporates effects of pressure gradients. Furthermore, the use of a wind-farm drag model which computes analytically the wake of each turbine would allow us to investigate separately the influence of wake effects and gravity waves on the optimal turbine set-points. This is work for future research."

- Line 224 "asses" should be "assess".

Thank you. We have corrected the erratum.

- Line 322, Figure 2: perhaps I missed it, but what height are the plots shown at? Perhaps include that in the caption.
  The three-layer model is a perturbation model which solves depth-averaged RANS equations. Therefore, we cannot visualize the flow field at different heights. Instead, we can show the perturbations values with respect to the background state. This is what Fig. 2 illustrates. We note that in the text we have used the words "top view", which could be misleading. Hence, we have modified the following sentence at P13-L350:

  "Figure 2 illustrates a planform view of the perturbation flow patterns obtained with $F_r = 0.9$ (top row) and $F_r = 1.1$ (bottom row) using the reference model setup."

- Overall, my thoughts are wondering how the results would change when you include changing wind direction into the problem. It seems that it would complicate things and maybe reduce the stability of the gravity waves in the atmosphere. I am also thinking that if I was a wind farm owner, why would I implement this? What are the implications on loads? How can the atmosphere be measured to provide input to the wind farm in order to modify individual turbine set points? etc. Of coarse, all of that is outside of the scope of this well written paper, but just thoughts to think of in taking this and making it into something practical.
  These are interesting questions. Indeed, the gravity wave system can become much more complex when multiple layers exist in the free atmosphere, introducing reflection of waves, as well as when baroclinic conditions appear. There is potentially a lot of research to be done to study all these effect. We are currently working on extending our gravity wave model to include such effects, and once available, they could also be included in optimization and control approaches.
  Determining the atmospheric state and wind direction (which is an input to the model) will be an additional challenge. This could be done using on-site lidars, and may be also based on weather forecasting, or weather radar. There is definitely still a lot of research to be done before our work can be applied in a real wind farm. In the manuscript, we have not further discussed on these issues, as they are still very speculative, and a first important step is to improve models and validation.